# Stratification of viral shedding patterns in saliva of COVID-19 patients

Hyeongki Park[1,2], Yoshimura Raiki[1], Shoya Iwanami[1], Kwangsu Kim[1,3,4], Keisuke Ejima[5,6], Naotoshi Nakamura[1], Kazuyuki Aihara[7], Yoshitsugu Miyazaki[8], Takashi Umeyama[8], Ken Miyazawa[8], Takeshi Morita[9], Koichi Watashi[9], Christopher B Brooke[10,11], Ruian Ke[12], Shingo Iwami[1,7,13,14,15,16,17]\*†, Taiga Miyazaki[18]\*†

[1]interdisciplinary Biology Laboratory (iBLab), Division of Natural Science, Graduate School of Science, Nagoya University, Nagoya, Japan; [2]School of Biomedical Convergence Engineering, Pusan National University, Yangsan, Republic of Korea; [3]Department of Science System Simulation, Pukyong National University, Busan, Republic of Korea; [4]Department of Mathematics, Pusan National University, Busan, Republic of Korea; [5]Lee Kong Chian School of Medicine, Nanyang Technological University, Singapore, Singapore; [6]The Tokyo Foundation for Policy Research, Tokyo, Japan; [7]International Research Center for Neurointelligence, The University of Tokyo Institutes for Advanced Study, The University of Tokyo, Tokyo, Japan; [8]Department of Chemotherapy and Mycoses, National Institute of Infectious Diseases, Tokyo, Japan; [9]Research Center for Drug and Vaccine Development, National Institute of Infectious Diseases, Tokyo, Japan; [10]Department of Microbiology, University of Illinois at Urbana-Champaign, Urbana, United States; [11]Department of Statistics, University of Illinois at Urbana-Champaign, Urbana, United States; [12]Theoretical Biology and Biophysics, Los Alamos National Laboratory, Los Alamos, United States; [13]Institute of Mathematics for Industry, Kyushu University, Fukuoka, Japan; [14]Institute for the Advanced Study of Human Biology (ASHBi), Kyoto University, Kyoto, Japan; [15]Interdisciplinary Theoretical and Mathematical Sciences Program (iTHEMS), RIKEN, Saitama, Japan; [16]NEXT-Ganken Program, Japanese Foundation for Cancer Research (JFCR), Tokyo, Japan; [17]Science Groove Inc., Fukuoka, Japan; [18]Division of Respirology, Rheumatology, Infectious Diseases, and Neurology, Department of Internal Medicine, Faculty of Medicine, University of Miyazaki, Miyazaki, Japan

**\*For correspondence:**
iwamishingo@gmail.com (SI);
taiga_miyazaki@med.miyazaki-u.ac.jp (TM)

†These authors contributed equally to this work

## eLife Assessment

This **important** work attempts to understand observed variability in oral shedding of SARS-CoV-2 and suggests that routine clinical factors are not determinative. The evidence supporting the conclusion is **solid** though the limited clinical heterogeneity of the included cohorts, the lack of COVID vaccination, and the absence of comprehensive viral load data for model training, makes the results difficult to generalize to contemporaneous COVID-19 conditions. This study may be of interest to virologists, public health officials and clinicians.

**Abstract** Living with COVID-19 requires continued vigilance against the spread and emergence of variants of concern (VOCs). Rapid and accurate saliva diagnostic testing, alongside basic public health responses, is a viable option contributing to effective transmission control. Nevertheless, our knowledge regarding the dynamics of SARS-CoV-2 infection in saliva is not as advanced as our

understanding of the respiratory tract. Here, we analyzed longitudinal viral load data of SARS-CoV-2 in saliva samples from 144 patients with mild COVID-19 (a combination of our collected data and published data). Using a mathematical model, we quantified individual-level viral dynamics and stratified them into three groups using a clustering approach. Notably, the three groups exhibited distinct differences in viral RNA detection durations: 11.5 days (95% CI: 10.6–12.4), 17.4 days (16.6–18.2), and 30.0 days (28.1–31.8), respectively. Surprisingly, this stratified grouping remained unexplained despite our analysis of 47 types of clinical data, including basic demographic information, clinical symptoms, results of blood tests, and vital signs. Additionally, we quantified the expression levels of 92 micro-RNAs in a subset of saliva samples, but these also failed to explain the observed stratification, although the mir-1846 level may have been weakly correlated with peak viral load. Our study provides insights into SARS-CoV-2 infection dynamics in saliva, highlighting the challenges in predicting the duration of viral RNA detection without indicators that directly reflect an individual's immune response, such as antibody induction. Given the significant individual heterogeneity in the kinetics of saliva viral shedding, identifying biomarker(s) for viral shedding patterns will be crucial for improving public health interventions in the era of living with COVID-19.

## Introduction

Coronavirus disease 2019 (COVID-19) vaccinations, which are effective in preventing infection by severe acute respiratory syndrome coronavirus 2 (SARS-CoV-2) and severe COVID-19 illness, have enabled the gradual and safe removal of COVID-19 restrictions on everyday life over the past year. However, we still face the emergence of variants of concern (VOCs), which is a major worry in the era of 'living with COVID-19'. As a result, to prevent major outbreaks, basic public health responses such as testing, isolation, and quarantine are demanded and important. In particular, rapid and accurate diagnostic tests are essential for controlling ongoing transmission. Salivary diagnostic testing is a convenient tool for early and efficient diagnosis of COVID-19 because it is easy for health care professionals and patients to administer (*Baghizadeh Fini, 2020*; *Wyllie et al., 2020*).

The oral cavity is an important target for SARS-CoV-2 (*Huang et al., 2021*), and viral particles in the lower and upper respiratory tract can reach the oral cavity through liquid droplets (*Baghizadeh Fini, 2020*). Saliva droplets are thus a potential route of SARS-CoV-2 transmission (*Baghizadeh Fini, 2020*; *Huang et al., 2021*). Although SARS-CoV-2 infection dynamics within the upper respiratory tract are well characterized thanks to data from oropharyngeal and nasopharyngeal swabs (*Kim et al., 2021*; *Ke et al., 2022*; *Jeong et al., 2022*; *Goyal et al., 2020*; *Marc et al., 2021*; *Néant et al., 2021*), infection dynamics in saliva are poorly understood. One potential explanation is the practical challenges inherent in using saliva as a specimen type. Handling steps such as tube opening, pipetting, and vortexing can generate infectious aerosols, requiring strict biosafety precautions (e.g. work in a biosafety cabinet). Also, salivary RNases and the abundant oral microbiota may degrade viral RNA or increase background noise, particularly in low viral load samples.

Recent reports (*Huang et al., 2021*; *Ke et al., 2022*) have suggested that SARS-CoV-2 infection dynamics differ qualitatively across tissues. Salivary gland tissue (parotid, submandibular, and sublingual) and other tissues are highly compartmentalized, and saliva represents the corresponding biofluid of the oral compartment. However, although individual-level heterogeneity in virus dynamics (especially individual infectiousness) has been evaluated, it remains unknown how individual-level viral shedding patterns in saliva are stratified, and which factor(s) determine the patterns (*Ke et al., 2022*; *Iwanami et al., 2021*). Since saliva is a route of direct viral transmission, salivary viral load provides particularly valuable information for assessing a patient's immediate infectiousness compared with measurements from other anatomical sites. Accordingly, to shape a country's early response to future VOCs, such as through isolation and screening guidelines based on salivary diagnostic testing, it is critical to address these points with the use of a high quality and quantity of saliva specimens annotated with basic clinical patient data.

So far, our ability to understand and characterize whole SARS-CoV-2 infection dynamics has been hindered by several limitations of the clinical data that made it impossible to capture either the early or the late phase of infection or to annotate individual-level clinical information. To overcome these limitations, here we quantified and stratified longitudinal virus dynamics in saliva samples from 144 mildly symptomatic participants of two different but complementary cohorts (*Ke et al., 2022*;

*Hosogaya et al., 2021*; *Miyazaki et al., 2023*). We successfully identified three groups with distinct viral shedding patterns. Notably, the groups showed clear differences in the duration of viral RNA detection (the mean durations were 11.5 days, 17.4 days, and 30.0 days, respectively). These findings imply a large inter-participant heterogeneity in virus infection dynamics. However, when we analyzed a total of 47 variables, including basic demographic information, daily clinical symptoms, results of blood tests, and vital signs, none explained the stratified grouping.

We also retrospectively explored whether salivary micro-RNAs were associated with the stratification by using stored residual saliva specimens. Micro-RNAs are non-coding RNAs that regulate numerous cellular processes by modulating protein levels through direct binding to mRNA (coding-RNA), thereby influencing translation efficiency or mRNA abundance. Various microRNAs are recognized for their impact on the viral replication ability and immune response in viruses like EBV, HCV, and HIV etc (*Trobaugh and Klimstra, 2017*). Many studies are underway to investigate micro-RNAs as potential targets for virus diagnosis and treatment. However, the relationship between micro-RNAs and the patterns of virus shedding of the SARS-CoV-2 virus in the body remains unknown. We quantified the expression levels of 92 micro-RNAs and found that no micro-RNA significantly explained the stratified groups, although the mir-1846 level may have been weakly correlated with the peak viral load. Our findings provide important insights into the complexities of viral shedding patterns in saliva and suggest that predicting the heterogeneity of viral dynamics using basic clinical and micro-RNA data may be a challenging task. These insights are critical for developing accurate diagnostic tools, effective treatments, and prevention strategies for COVID-19.

## Results
### Description of cohort and study design
We used longitudinal saliva viral load data obtained from cohorts of the nelfinavir (NFV) clinical trial (jRCT2071200023 *Hosogaya et al., 2021*; *Miyazaki et al., 2023*) and the University of Illinois at

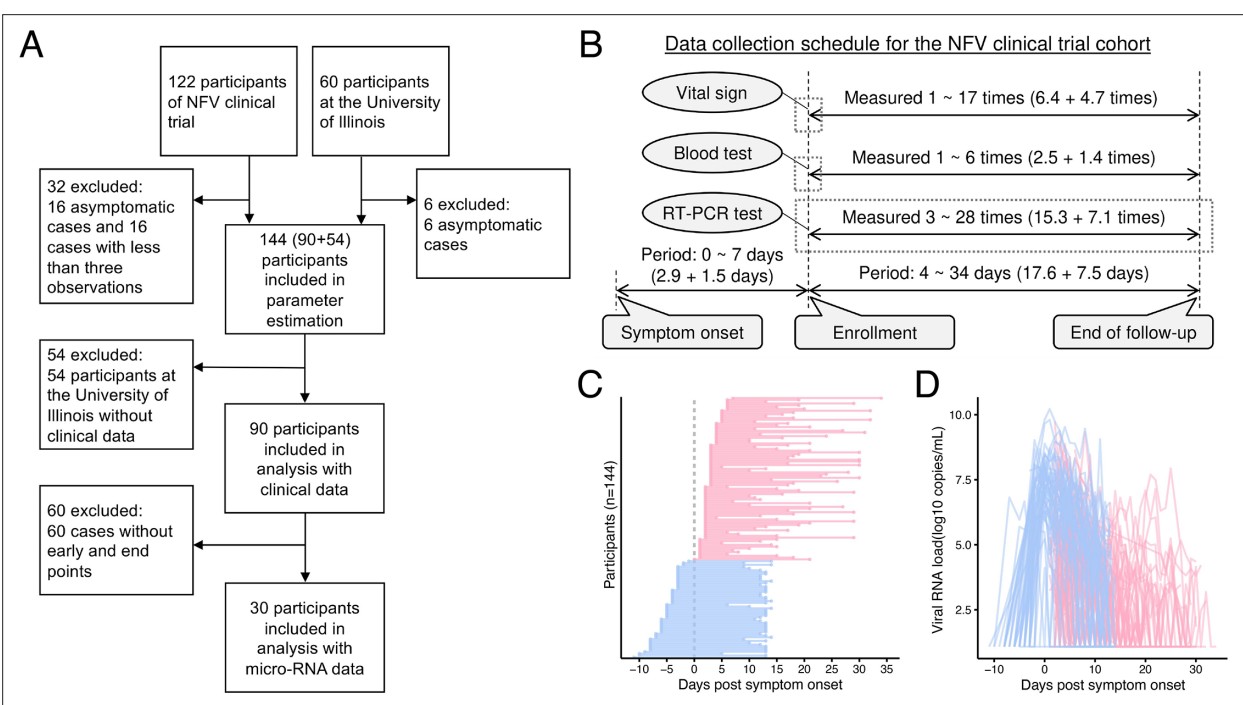

**Figure 1.** Characteristics of cohorts from the NFV clinical trial (jRCT2071200023 *Hosogaya et al., 2021*; *Miyazaki et al., 2023*) and the University of Illinois (*Ke et al., 2022*). (**A**) Flowchart of cohorts from the NFV clinical trial and the University of Illinois along with the number of participants and inclusion criteria for our analysis is described. (**B**) Data collection schedule of viral load, blood test, and vital signs from participants in the NFV clinical trial is described. The gray box highlights the measurement dates for the observations included in the analysis. (**C**) and (**D**) show, for each participant (N=144 participants, 2191 samples), the timeline of sample collection and the captured SARS-CoV-2 viral RNA load for saliva RT-qPCR, respectively. The red and blue colors indicate samples for cohorts from the NFV clinical trial and the University of Illinois, respectively.

Urbana-Champaign (*Ke et al., 2022*). All infections were either mild or asymptomatic. All participants in these cohorts reported through surveys (self-declaration) that they had never been previously infected with SARS-CoV-2, and none were vaccinated against SARS-CoV-2 at the time of enrollment (see Materials and methods for details, including the measurement dates). Of 182 participants from these two studies, 144 symptomatic participants, excluding 22 asymptomatic participants and 16 participants with incomplete observational data, were considered in our analysis (*Figure 1A*). In addition, we annotated the saliva viral load results of 90 participants from the NFV clinical trial with their sex, age, daily symptoms, blood test results, and vital signs. These clinical data were collected longitudinally for each patient (*Figure 1B*). While all patients had clinical measurements at the enrollment time point (a mean of 2.9 days after symptom onset), the frequency of subsequent measurements varied among individuals (*Figure 1B*). Therefore, our analysis focused on the clinical data obtained at enrollment, which were consistently available across all patients. This approach is also appropriate given the study's aim of identifying early predictors of viral dynamics (see later).

*Table 1* and *Supplementary file 1A* present the mean values of clinical information and daily symptom data, respectively. The clinical data described in *Table 1* were collected only for the NFV cohort. Therefore, we first used both cohorts to estimate viral dynamics, but in the subsequent analyses examining the relationship between the clinical data and estimated viral dynamics, we relied only on the NFV cohort. Because daily symptom data described in *Supplementary file 1A* were collected for both cohorts, both cohorts were included in the analyses relating daily symptom data to viral dynamics.

The data from the two studies complemented one another: the longitudinal data from the University of Illinois contained data on the early phase of infection (before symptom onset), while the data from the NFV clinical trial contained data on the late phase of viral RNA load (more than 14 days after symptom onset) (*Figure 1C*). Viral load data were collected an average of 15.2 times per participant (SD, 5.7). Pre-symptom onset measurements averaged 0 (SD, 0) in the NFV cohort and 3 (SD, 1) in the Illinois cohort. At 14 or more days after symptom onset, the corresponding averages were 5.5 (SD, 6.1) and 0.1 (SD, 0.3), respectively. The time-series viral load in the saliva samples for all individuals used in the analysis is plotted in *Figure 1D*. We also analyzed 60 stored saliva specimens from 30 participants of the NFV clinical trial for micro-RNA analysis.

## Quantifying and stratifying SARS-CoV-2 infection dynamics in saliva

We employed a previously developed mathematical model describing SARS-CoV-2 infection dynamics (i.e. *Equations 1-2*) to evaluate interparticipant heterogeneity (details are provided in Materials and methods), and reconstructed the best-fit virus dynamics in saliva of 144 symptomatic participants (*Figure 2—figure supplement 1* and *Supplementary file 1B*). We also applied a mechanistically more realistic mathematical model *Equations 3-6* that we had previously used with the Illinois dataset (*Ke et al., 2022*; *Supplementary file 1C*). Both models showed stable results in the visual predictive checks (VPC) and convergence assessments and adequately described the data (*Figure 2—figure supplement 2* and *Figure 2—figure supplement 3*). The fit with *Equations 3-6* yielded a lower Akaike Information Criterion (AIC) of 6720, compared with 6834 obtained with *Equations 1-2*. Nevertheless, the reconstructed viral dynamics in the NFV clinical trial dataset were highly similar between the two models, whereas somewhat larger differences were observed for the Illinois dataset (*Figure 2—figure supplement 4*). This may reflect the inclusion of late- and end-phase viral load measurements in the NFV trial, which were not available in the Illinois dataset. It should be noted that the primary aim of applying mathematical modeling in this study was not to dissect viral mechanisms in detail, but to reconstruct viral dynamics for stratification purposes. Accordingly, we adopted the simpler model (i.e. *Equations 1-2*), which yields comparable dynamics while being more practical and sufficient for reconstructing viral load in this context. This approach also avoids the additional complexity and parameter assumptions required by *Equations 3-6*, as described in *Ke et al., 2022*.

Next, to stratify the time-course pattern of viral shedding, we first applied unsupervised random forest clustering to the individual 'reconstructed' virus dynamics of 144 participants (e.g. *Figure 2—figure supplement 1*). However, this analysis failed to divide the time-course pattern into different clusters (data not shown). To overcome this problem, we quantified the peak, duration, up-slope (i.e. growth rate), and down-slope (i.e. decay rate) of the reconstructed dynamics as 'features' of the virus dynamics (*Supplementary file 1D*). Interestingly, the unsupervised random forest clustering based on

**Table 1.** Clinical data of the overall cohort from the NFV clinical trial and in groups stratified by longitudinal virus dynamics.

| Clinical data | Unit | Group1 (N=33) | Group2 (N=37) | Group3 (N=20) | Overall (N=90) |
|---|---|---|---|---|---|
| **Basic demographic information** | | | | | |
| Age | years | 47 (13.9)* | 40.1 (12.6) | 43.4 (13.7) | 43.7 (13.6) |
| Sex ‡ | - | 46% | 67% | 70% | 59% |
| **Vital signs** | | | | | |
| Systolic blood pressure | $mmHg$ | 116 (16.3) | 119.8 (14.4) | 122.2 (20.3) | 118.8 (16.6) |
| Diastolic blood pressure | $mmHg$ | 77.5 (14.3) | 75.9 (10.9) | 78.5 (14.8) | 77.1 (13.2) |
| Pulse rate | $bpm$ | 83.5 (16.9) | 86.9 (14.3) | 79.1 (13.1) | 83.8 (15.3) |
| SpO$_2$ | % | 97.5 (1) | 97.7 (0.8) | 97.6 (0.9) | 97.6 (0.9) |
| Respiratory rate | $bpm$ | 17 (3.9) | 16.5 (3.4) | 16.8 (2.4) | 16.8 (3.4) |
| **Blood test results** | | | | | |
| White blood cell count | $10^9/L$ | 5.1 [30.2]† | 5.3 [31.9] | 5.9 [34] | 5.3 [32.8] |
| Neutrophil | % | 62.4 (10.3) | 60 (11.1) | 59.8 (11.4) | 61 (10.8) |
| Eosinophil | % | 1 (1.8) | 0.9 (1.5) | 1 (1.2) | 1 (1.6) |
| Basophil | % | 0.3 (0.2) | 0.4 (0.3) | 0.4 (0.3) | 0.3 (0.3) |
| Lymphocytes | % | 26.1 (8.4) | 28.1 (10) | 28 (9.2) | 27.3 (9.1) |
| Monocyte | % | 10.1 (3.7) | 10.6 (3.5) | 10.8 (4.2) | 10.4 (3.7) |
| Red blood cell count | $10^{12}/L$ | 5 [450.3] | 5.5 [464.1] | 5.4 [450.9] | 5.3 [453] |
| Amount of hemoglobin | $g/dL$ | 14 (1.4) | 15.1 (1.1) | 14.8 (1.3) | 14.6 (1.4) |
| Hematocrit | % | 41.6 (3.6) | 44.1 (3.2) | 43.8 (3.6) | 43 (3.6) |
| Platelet count | $10^9/L$ | 121.6 (88.7) | 127.8 (98) | 136.6 (102.4) | 127.2 (94.4) |
| CRP | $mg/dL$ | 1.3 (2.6) | 1.3 (1.9) | 1.1 (1.8) | 1.3 (2.2) |
| Protein | $g/dL$ | 7.4 (0.4) | 7.4 (0.4) | 7.4 (0.3) | 7.4 (0.4) |
| Albumin | $g/dL$ | 4.2 (0.3) | 4.3 (0.3) | 4.4 (0.3) | 4.3 (0.3) |
| ALT | $U/L$ | 20.2 (11.9) | 29.9 (23.9) | 31.3 (22) | 26.2 (19.8) |
| AST | $U/L$ | 21.9 (7.8) | 28.3 (14) | 28.7 (13.4) | 25.8 (12) |
| γ-GTP | $U/L$ | 30.9 (25.6) | 42 (41.8) | 47.8 (35) | 38.7 (34.7) |
| ALP | $U/L$ | 69.9 (20.9) | 72 (20.3) | 72.8 (20.6) | 71.3 (20.4) |
| LDH | $U/L$ | 172.1 (34.1) | 184.1 (43.8) | 186.6 (29.5) | 179.7 (37.2) |
| Bilirubin | $mg/dL$ | 0.5 (0.2) | 0.5 (0.2) | 0.6 (0.2) | 0.5 (0.2) |
| CK | $U/L$ | 85.2 (48.2) | 126.7 (146.5) | 118.2 (133) | 107.7 (113.1) |
| Na | $mEq/L$ | 139.3 (2.9) | 139.9 (2.2) | 139.8 (2.8) | 139.6 (2.6) |
| K | $mEq/L$ | 3.9 (0.3) | 3.9 (0.3) | 3.9 (0.2) | 3.9 (0.3) |
| Cl | $mEq/L$ | 102.6 (3.3) | 102.7 (3.6) | 102.8 (2.7) | 102.6 (3.3) |
| BUN | $mg/dL$ | 11.4 (3) | 11.3 (2.5) | 12.5 (3.1) | 11.6 (2.9) |
| CRE | $mg/dL$ | 0.8 (0.2) | 0.8 (0.2) | 0.9 (0.2) | 0.8 (0.2) |
| Blood glucose | $mg/dL$ | 105.9 (19.9) | 102.4 (17.1) | 109 (15.7) | 105.3 (18) |

*Table 1 continued on next page*

*Table 1 continued*

| Clinical data | Unit | Group1 (N=33) | Group2 (N=37) | Group3 (N=20) | Overall (N=90) |
|---|---|---|---|---|---|
| HbA1c | % | 5.6 (0.3) | 5.5 (0.4) | 5.6 (0.3) | 5.6 (0.3) |
| Procalcitonin | *μg/L* | 0.1 (0.2) | 0.1 (0.1) | 0.1 (0.04) | 0.1 (0.1) |
| Fibrinogen | *mg/dL* | 371 (94.8) | 365.3 (101.1) | 2 (73) | 364.1 (92.3) |
| PT | % | 103.1 (20.5) | 105.5 (17.5) | 109.3 (15.5) | 105.4 (18.4) |
| APTT | *sec* | 31.4 (3.7) | 31.1 (5.3) | 30.7 (3.2) | 31.1 (4.2) |
| PT-INR | - | 1 (0.1) | 1 (0.1) | 1 (0.1) | 1 (0.1) |
| D-dimer | *mg/L* | 0.8 (1.3) | 0.5 (0.2) | 0.5 (0.1) | 0.6 (0.8) |

*Mean (standard deviation).

†median [Interquartile range].

‡Sex was reported as the proportion of males.

these features identified three groups (i.e. G1: N=46, G2: N=61, and G3: N=37) in which the time-course patterns were clearly discriminated. This finding suggested that there is a heterogeneity of virus infection dynamics in saliva (see Materials and methods). *Figure 2A* indicates a two-dimensional Uniform Manifold Approximation and Projection (UMAP) embedding of the three stratified groups. Using a different color for each group (gray for G1, magenta for G2, and blue for G3), we also plotted the estimated individual viral load (*Figure 2B*), and the highlighted time-course pattern of each group by the Partial Least-Squares Discriminant Analysis (PLS-DA; *Figure 2C*). In addition, we evaluated the impact of statistical uncertainty in the estimated viral dynamics on the stratification by comparing the uncertainty-adjusted and original distance matrices using the Mantel test (see Materials and methods). The two matrices were strongly correlated (Mantel $r$=0.72, p<0.001), indicating that the stratification is robust to parameter uncertainty.

The distributions of the four features are described in *Figure 2D*; a statistically significant between-group difference was found in the duration of viral RNA detection. The mean durations were 11.5 days (95% CI: 10.6–12.4), 17.4 days (16.6–18.2), and 30.0 days (28.1–31.8) for G1, G2, and G3, respectively. In our previous report (*Iwanami et al., 2021*), we consistently confirmed that there were at least 3 groups showing different duration of viral RNA detection in upper respiratory specimens.

Because of previous work concluding that there are no significant differences in virus infection dynamics between NFV-treated and untreated participants (*Miyazaki et al., 2023*), we analyzed all data together regardless of treatment (see Materials and methods). To further confirm whether NFV affects the stratification of the time-course pattern of viral shedding, we compared the number of individuals belonging to each group (i.e. G1, G2, and G3) between NFV-treated and untreated participants (including the members of the University of Illinois cohort) and found no trend for the stratification (p=0.784 by the Fisher's exact test: *Figure 2E*).

Another possibility that explains the different viral duration observed here may be a difference in VOC genotypes. To test this, we used data from 55 participants in our NFV clinical trial who had been characterized according to which VOCs (i.e. B.1.1.7 [Alpha], B.1.672.2 or AY.29 [Delta], and other variants) they had been infected with (see Materials and methods), in addition to data from 54 participants of the University of Illinois cohort. However, we observed no trend in the number of individuals belonging to each group among the VOCs (p=0.728 by the Fisher's exact test: *Figure 2F*), which is consistent with the conclusion in *Ke et al., 2022*.

## Basic clinical data may not explain heterogeneity in individual viral shedding

Using data from the NFV clinical trial, we annotated the saliva viral loads of 90 participants with basic demographic information, daily symptoms, blood test results, and vital signs (*Figure 1A*, *Table 1* and *Supplementary file 1A*). We also annotated the saliva viral loads of 52 participants from the University of Illinois with daily symptoms (*Supplementary file 1A*).

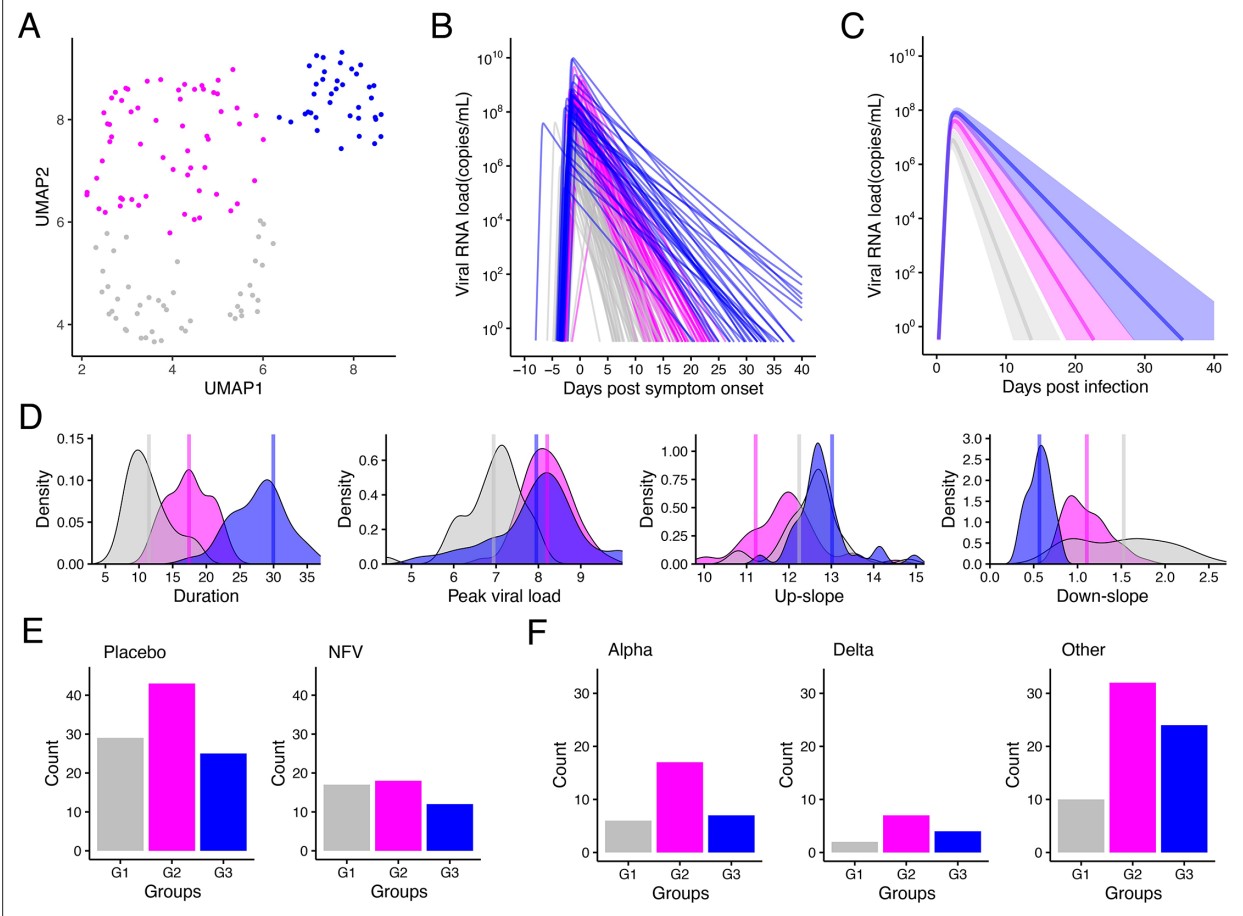

**Figure 2.** Stratification of individual SARS-CoV-2 viral dynamics in saliva. (**A**) UMAP of stratified viral RNA load based on the extracted features from the reconstructed individual-level viral dynamics is shown. (**B**) The reconstructed individual viral RNA load is shown. Colors for individual-level viral dynamics correspond to the colors of the dots in the UMAP described in (**A**). (**C**) The time-course patterns of each group highlighted by the Partial Least-Squares Discriminant Analysis (PLS-DA). (**D**) Distributions between groups of each feature used for stratification of viral shedding patterns are shown. The p-values of ANOVA for the difference in each feature among stratified groups are all less than 0.05. (**E**) Distributions of the number of individuals in each stratified group for the standard-of-care alone (left, n=97) and standard-of-care plus NFV administration (right, n=47) participants are shown. (**F**) Distributions of the number of individuals in each stratified group for Alpha variants (left, n=30), Delta variants (middle, n=13), and other variants (right, n=66) of SARS-CoV-2 are shown.

The online version of this article includes the following figure supplement(s) for figure 2:

**Figure supplement 1.** Reconstructed viral dynamics in saliva samples for individual participants.

**Figure supplement 2.** Visual predictive checks (VPC) for the viral load models.

**Figure supplement 3.** Convergence diagnostics of the SAEM algorithm.

**Figure supplement 4.** Comparison of three model fits to viral load in saliva samples for individual participants.

**Figure supplement 5.** Sensitivity analysis of the detection limit for viral load data in the NFV cohort.

To identify factors that were significantly correlated with the viral shedding patterns in saliva specimens obtained from the NFV clinical trial, we first examined the 39 variables summarized in *Table 1*. Each factor was compared between the three groups by ANOVA, and the p-values were corrected by the False Discovery Rate (FDR). However, we found no clinical data that differed significantly (i.e., corrected p-value of ANOVA of less than 0.05) among the stratified groups (*Figure 3A*). To avoid overfitting by bootstrap aggregating (bagging), we also trained a random forest classifier (see Materials and methods), a tree-based machine learning algorithm suitable for tabular data (*Grinsztajn et al., 2022*), to predict the group from the clinical data of 90 individuals in the NFV clinical trial cohort and obtained ROC-AUCs of 60%, 49%, and 36% for predicting G1, G2, and G3, respectively (*Figure 3B*). We were not able to achieve a high ROC-AUC for predicting the shedding patterns based on the basic

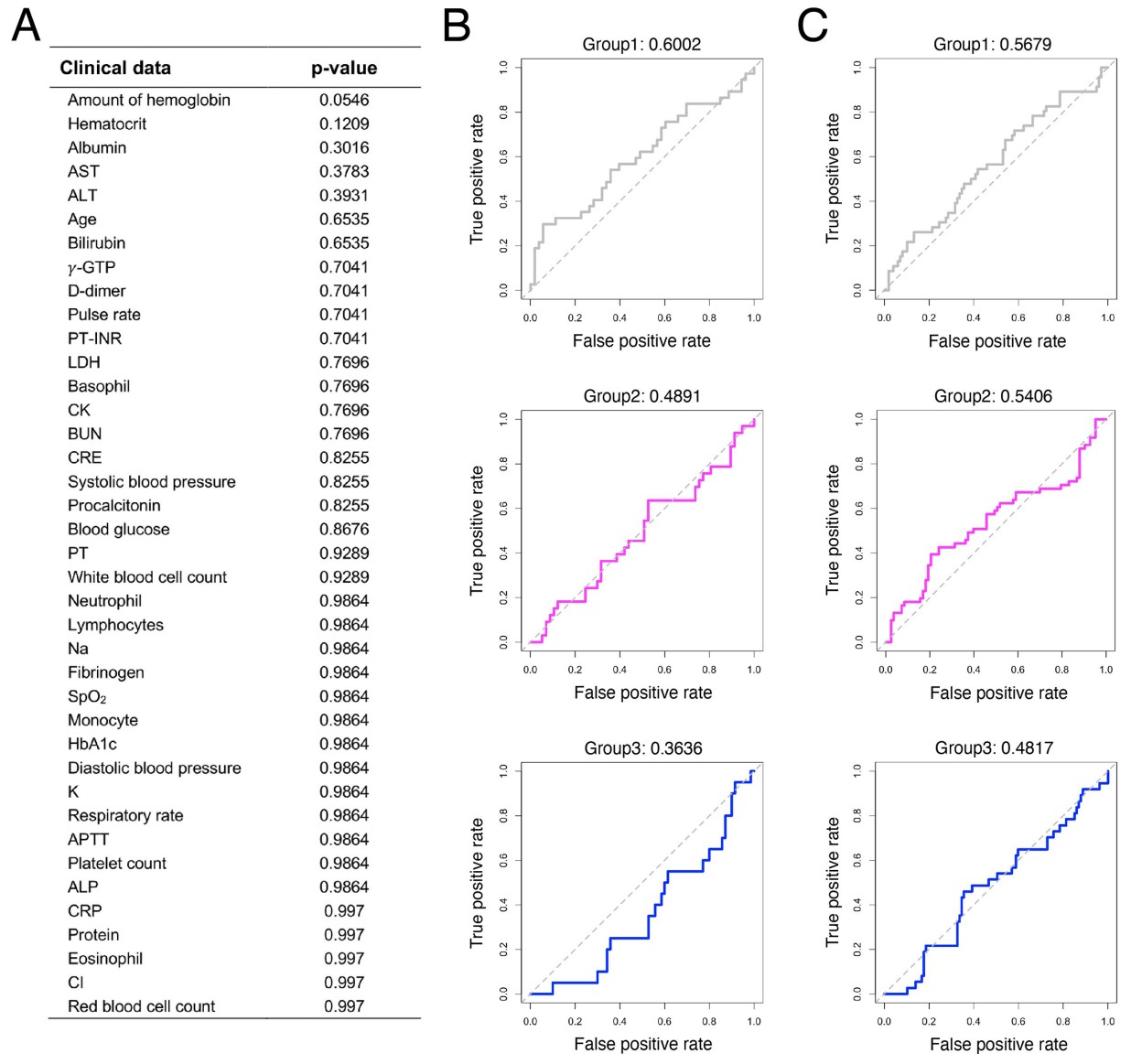

**Figure 3.** Correlation between clinical data and viral shedding patterns. (**A**) p-Values of ANOVA corrected by the FDR to compare clinical data among the three stratified groups are shown. Clinical data are listed in reverse order of p-values. (**B**) and (**C**) show ROC curves of random forest classifiers trained on predicting each group by using data for (**B**) clinical values described in *Table 1* and (**C**) symptom onset data described in *Supplementary file 1A*, respectively. The corresponding AUC (area under curve) value of each ROC curve is shown on the top of each panel.

clinical data. We also attempted to make predictions based on clinical data which exhibited relatively low p-values in the ANOVA analysis. However, we were unable to achieve a high prediction accuracy with this approach (data not shown).

Next, we asked whether the stratification of the study population is associated with clinical symptoms of COVID-19 that could be caused by active replication of SARS-CoV-2. In general, the clinical symptoms of COVID-19 are cough, fever, shortness of breath, muscle pain, sore throat, confusion, chest pain, headache, rhinorrhea, diarrhea, and nausea and vomiting. In our study, individual-level symptom data were available as eight categories in the cohorts from both the NFV clinical trial and the University of Illinois (*Supplementary file 1A*). Symptom data were obtained via participant self-report and encoded as categorical indicators (presence/absence) for each symptom. We tried to use a random forest classifier to investigate whether symptom data could predict the stratified groups and obtained ROC-AUCs of 57%, 54%, and 48% for predicting G1, G2, and G3, respectively (*Figure 3C*).

In fact, SARS-CoV-2 human challenge clearly showed no quantitative correlation between the individuals' time-series pattern of viral load and symptoms (*Killingley et al., 2022*).

Additionally, we investigated the relationship between each feature of viral dynamics (i.e. duration of viral RNA detection, peak viral load, up-slope, and down-slope) and the clinical data by using the Pearson's correlation coefficient (*Supplementary file 1E*). Overall, correlation coefficients were low (0.06 on average) with high p-values, which suggests that no feature was likely to be explained by these clinical data.

## Relationship between salivary micro-RNAs and viral shedding patterns in COVID-19 patients

Various proteins in saliva have antiviral effects. It is also expected that some micro-RNAs in saliva may impair SARS-CoV-2 replication (*Baghizadeh Fini, 2020*). In addition, saliva-based micro-RNAs have already been reported to be associated with various diseases (*Alizadeh et al., 2025*; *Domínguez-de-Barros et al., 2025*; *Setti et al., 2020*; *Xie et al., 2015*; *Zhang et al., 2024*; *Pimenta et al., 2021*), including COVID-19 (*Ahamed et al., 2025*). In particular, although the direct relationship between micro-RNA expression levels and viral load in the body has not yet been clarified, several studies have reported that micro-RNAs are associated with the severity of COVID-19 (*Hicks et al., 2023*; *Saulle et al., 2023*). Based on this, we anticipated that among the biomarkers obtainable from saliva, micro-RNAs could potentially serve as non-invasive predictors of viral dynamics. Accordingly, despite being less readily obtainable than clinical information, micro-RNAs are accessible from saliva samples and were anticipated to be closely linked to viral dynamics, so we incorporated them into our analysis.

We here used the stored residual saliva specimens from the NFV clinical trial to identify micro-RNA(s) associated with the stratified groups (i.e. G1, G2, and G3). We note that, because all residual

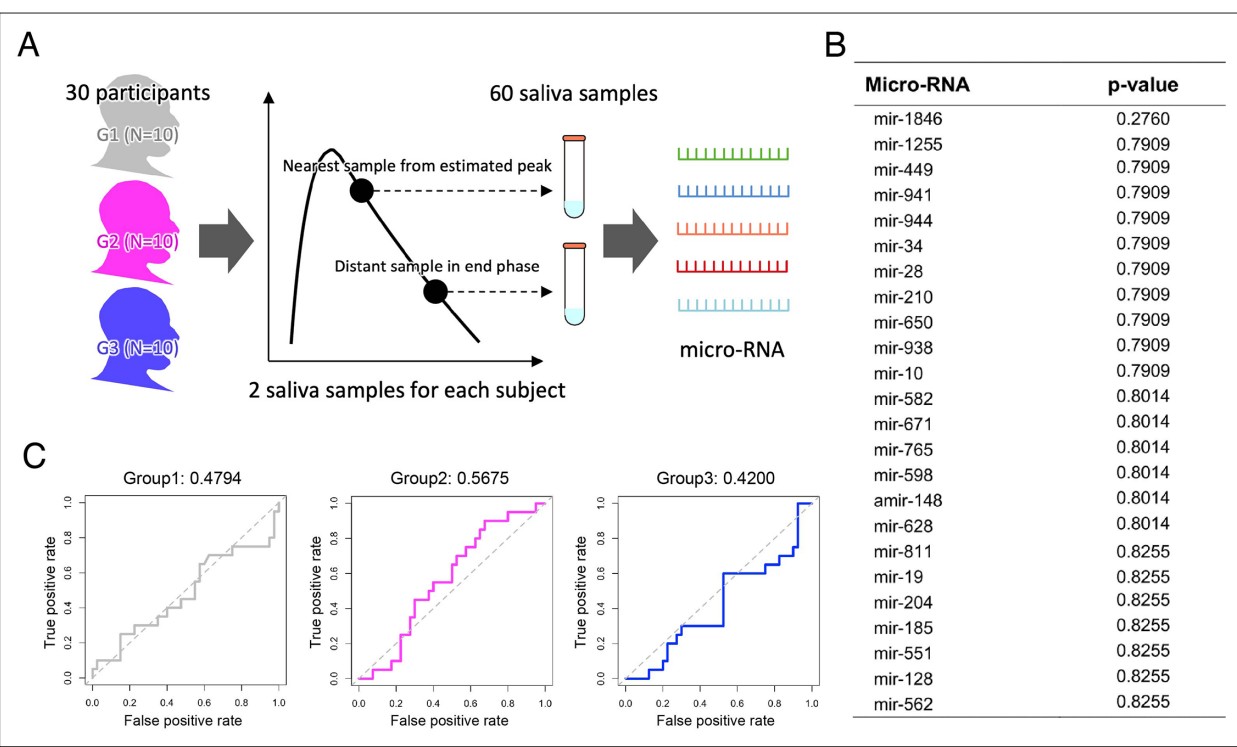

**Figure 4.** Correlation between micro-RNA data and viral shedding patterns. (**A**) The strategy of micro-RNA data collection from saliva samples in the NFV clinical trial is described. We picked a total of 30 participants by choosing 10 participants from each group. We chose two samples (the nearest to estimated peak and the most distant but above the detection limit in the late phase) from each participant for quantifying micro-RNA. (**B**) p-Values of Kruskal-Wallis ANOVA corrected by the FDR for each micro-RNA level are shown. Micro-RNA levels are listed by reverse order of p-values. Only the 24 micro-RNA levels with the lowest p-values are shown. (**C**) ROC curves of random forest classifiers trained on predicting each group by using levels of 92 micro-RNAs are shown. The corresponding AUC value of each ROC curve is presented on the top of each panel.

The online version of this article includes the following figure supplement(s) for figure 4:

**Figure supplement 1.** Correlation between mir-1846 level and the features of infection dynamics.

saliva specimens are annotated with the individual participant and we know which participants belong to which stratified group, we can select and compare saliva specimens from G1, G2, and G3 without bias. This implies that we can impartially select participants in equal numbers from each group, unaffected by other factors. Specifically, we collected 60 stored saliva specimens from the NFV clinical trial to perform micro-RNA analysis for 30 participants. We picked two samples for each participant to evaluate the role of micro-RNAs during both the peak and the late phase (i.e. 30 samples for each phase): the nearest sample from the estimated peak and the most distant sample above the detection limit in the late phase (*Figure 4A*). We normalized micro-RNA expression among the samples by using the DESeq2 Bioconductor package. We summarize the information on the micro-RNAs we obtained from saliva specimens in *Supplementary file 1F*.

First, we compared micro-RNA levels between two phases using pairwise t-tests (and Mann-Whitney U tests) with FDR correction. As a result, no micro-RNA showed a statistically significant difference. This suggests that micro-RNA levels remain relatively stable during the course of infection and may therefore serve as a biomarker independent of sampling time.

Next, similar to the analysis using clinical data, we compared the expression levels of 92 micro-RNAs between the three stratified groups. Because the micro-RNA data were non-parametric, we used the Kruskal-Wallis ANOVA for analysis and corrected the p-values by FDR. However, we failed to find micro-RNA levels that differentiated the stratified groups (i.e. with a corrected p-value of Kruskal-Wallis ANOVA of less than 0.05) in the three trials using the data from the peak phase, the late phase, and both phases (e.g. *Figure 3B* for the total 60 samples). We also trained a random forest classifier to predict each group from the micro-RNA levels for the 60 total samples and obtained ROC-AUCs of 48%, 57%, and 42%, respectively. Again, we did not obtain enough ROC-AUCs to predict stratified groups by using the collected micro-RNA data.

Furthermore, we investigated the relationship between the four features of viral dynamics and micro-RNA levels. Here we used the Spearman's correlation coefficient (*Supplementary file 1G*). Overall, we did not find strong correlations between micro-RNA levels and features (Spearman's correlation coefficients on average of 0.002, 0.024, –0.001, and –0.001 for duration of viral RNA detection, peak viral load, up-slope, and down-slope, respectively, for the 60 total samples). Only the mir-1846 level exhibited a weak negative correlation (–0.53 Spearman's correlation coefficient with 0.01 p-value) with the peak viral load (*Figure 4—figure supplement 1*). We confirmed similar trends even when we analyzed the micro-RNA level for the peak and late phases separately.

## Discussion

Being able to quickly and efficiently diagnose COVID-19 is essential in monitoring the pandemic. Because the sampling process for saliva is noninvasive, and because it is inexpensive and minimizes the risk for transmissions to health care workers (*Baghizadeh Fini, 2020*), saliva sampling has excellent potential and advantages over other sampling methods from biological specimens such as the lower and upper respiratory tract (*Wyllie et al., 2020*; *To et al., 2020*). Given the significant individual heterogeneity in the saliva viral shedding (*Ke et al., 2022*; *Hay et al., 2022*), identifying biomarker(s) for viral shedding patterns will be crucial for improving public health interventions in the era of living with COVID-19.

To improve our understanding of SARS-CoV-2 infection dynamics in saliva to enable application of saliva testing in the fight against COVID-19, we quantified and stratified longitudinal virus dynamics in saliva samples from 144 mildly symptomatic individuals from the cohorts of the NFV clinical trial (*Hosogaya et al., 2021*) and the University of Illinois at Urbana-Champaign (*Ke et al., 2022*), and we uniquely analyzed the relationships among viral dynamics, clinical data, and micro-RNAs. Our mathematical modeling analysis indicates that viral dynamics in saliva may exhibit distinct patterns compared to those in the upper respiratory tract. We estimated that viral replication in saliva is characterized by a relatively rapid early growth phase, with a mean (standard deviation) doubling time of 1.44 (0.49) hours. Compared with prior studies analyzing viral dynamics in the upper respiratory tract using similar models, which reported doubling times of 2–4 hr (*Ke et al., 2022*; *Gunawardana et al., 2022*; *Iyaniwura et al., 2024*), our findings suggest that viral replication in saliva proceeds faster than in the upper respiratory tract. Multiple previous studies have also shown that viral loads in saliva rise more rapidly than in the nasal cavity, are detected with higher sensitivity early in infection, and reach

their peak earlier (*Ke et al., 2022*; *Migueres et al., 2022*; *Puhach et al., 2023*; *Savela et al., 2022*; *Smith et al., 2021*).

In addition to the large heterogeneity in virus infection dynamics, we identified three groups (i.e. G1, G2, and G3) with different viral shedding patterns (*Figure 2D*). Immunocompromised patients have been reported to have a prolonged duration of viral RNA detection, lasting over three months, underscoring the critical role of host immune responses in controlling viral infections (*Leung et al., 2022*; *Niyonkuru et al., 2021*; *Nakajima et al., 2021*; *Wei et al., 2021*). Although oral immune responses remain poorly understood, Huang et al. recently confirmed by using single-cell RNA sequencing of the human minor salivary glands and gingiva that SARS-CoV-2 infection can trigger sustained, localized immune responses in saliva (*Huang et al., 2021*). In this study, we observed significant differences in the down-slopes of viral shedding in saliva among participants in different groups, with a more rapid decline in G1. This decline is likely attributed to a stronger immune response to SARS-CoV-2 in G1 participants than in participants in G2 and G3, as reflected in the death rate of infected cells due to the immune response (*Figure 2D*). Lower levels of viral replication have also been observed among infected participants with high baseline levels of mucosal IgA (but not IgG), as reported elsewhere (*Havervall et al., 2022*). Recently, we demonstrated that rapid anti-spike secretory IgA antibody responses can contribute to reducing duration of viral RNA detection and amounts in nasopharyngeal mucosa (*Miyamoto et al., 2023*). These findings highlight the importance of biomarkers that directly reflect an individual's immune response, such as virus-specific antibody induction and T cell levels etc., in predicting viral shedding patterns. Therefore, quantifying the time-series pattern of mucosal IgA and its correlation with saliva viral load may provide crucial insights into the stratification of SARS-CoV-2 infection dynamics.

For the purpose of predicting viral shedding patterns during the early stage of infection, we first explored the association of 39 basic clinical variables, 8 daily symptoms, and the levels of 92 micro-RNAs with the stratified groups. However, none of the factors were significant (*Table 1*, *Figure 3A*, *Figure 4B*, *Supplementary file 1A* and *Supplementary file 1F*). On the other hand, it is noteworthy that all infection cases were mild and that most participants had clinical indicators within normal ranges. This lack of clinical heterogeneity within the cohort may have limited the ability to fully capture the diversity of infection dynamics. Moreover, the clinical parameters analyzed in this study are, a priori, unlikely to exhibit strong correlations with virologic outcomes. In contrast, we showed that mir-1846, which is an exogenous micro-RNA that is specifically classified as an Oryza sativa micro-RNA (osa-microRNA; *Rakhmetullina et al., 2020*), may exhibit a weak negative correlation. Exogenous micro-RNAs enter the human body primarily through food and can affect human metabolism by interacting and binding with human genes. mir-1846 is reported to interact with two human genes (*Rakhmetullina et al., 2020*) that are known to be associated with the progression of melanoma, various cancers, and leukemia. This suggests that mir-1846 levels may be linked to human immunity. Few studies have investigated the role of mir-1846 in humans, but our findings suggest the need for further investigations into the impact of this micro-RNA level on human immunity. Our research sheds light on the intricate patterns of viral shedding in saliva.

Our approach has several limitations that must be considered in our next study: First, our analysis was limited to participants with symptomatic infection and excluded those with asymptomatic infection (22 asymptomatic individuals out of a total of 182 individuals, i.e. 12% of participants) because we integrated datasets with different time scales from different cohorts. Although our data do not include participants infected with omicron variants, others have reported that the omicron variant may cause a higher proportion of asymptomatic infection (*Garrett et al., 2022*). Thus, evaluating the effect of asymptomatic infection will be important to update our stratification, especially for recent (or future emerging) VOCs. Second, potential viral rebound was neither prespecified nor systematically assessed. A subset of participants exhibited patterns consistent with possible rebound (e.g. S01-16 and S01-43 in *Figure 2—figure supplement 1*), which could affect estimates of viral RNA detection duration. Future studies should predefine an operational criterion for viral rebound and explicitly incorporate it into the modeling framework to strengthen robustness. Since both models considered in the present study cannot account for viral rebound, a more complex model would be required to capture this phenomenon. Third, micro-RNAs participate in the post-transcriptional regulation of gene expression; however, they do not provide direct insights into immune cell dynamics. Given the reported association between the duration of viral RNA detection

and mucosal immunity as discussed above, it appears imperative to analyze modalities that are directly linked to the immune response in the future. Fourth, some of our results may have limited relevance to the current COVID-19 situation, as most people have now either been infected or vaccinated. Nevertheless, investigating the relationship between viral shedding patterns in saliva and various clinical and microRNA data, and developing a method to do so, remains important. Such research can offer valuable insights into the early response to emerging infectious viruses in the future.

Another potential limitation of this study is the timing of saliva specimen sampling, although we took great care to select and compare specimens from G1, G2, and G3 without bias. As a result of our clinical trial design (jRCT2071200023 *Hosogaya et al., 2021*; *Miyazaki et al., 2023*), participants were enrolled after the onset of symptoms, thereby restricting saliva specimen collection exclusively to the post-symptom phase. Unfortunately, we lack samples from the pre-infection, pre-symptomatic, and early infection phases. Consequently, the absence of individual-level baseline values for micro-RNA means that inter-participant heterogeneity in micro-RNA levels may obscure signals related to distinct viral infection dynamics in saliva.

In conclusion, our study revealed that the dynamics of SARS-CoV-2 infection in saliva can be classified into three groups based mainly on the duration of viral RNA detection. However, accurately predicting the variability in viral dynamics remains a challenging task, because it requires a more comprehensive understanding of the complex shedding patterns in saliva, as well as detailed clinical and molecular data. The identification of a sensitive, simple, and rapid biomarker for saliva viral shedding will be imperative for future COVID-19 outbreak control.

# Materials and methods
## Saliva viral load data

Longitudinal saliva viral load data of participants with symptomatic and asymptomatic COVID-19 (122 cases) were obtained from the NFV clinical trial (*Hosogaya et al., 2021*). Briefly, the NFV clinical trial was a prospective, randomized, open-label, blinded-endpoint, parallel-group trial conducted between July 2020 and October 2021 at 11 university and teaching hospitals in Japan. This study consisted of a 14-day treatment period and a 14-day follow-up period, with no significant differences in the time to viral clearance between patients who received standard-of-care plus NFV administration and those who had the standard-of-care alone (*Hosogaya et al., 2021*; *Miyazaki et al., 2023*). Therefore, the participants with COVID-19 were analyzed together here. In addition, we obtained similar saliva viral load data (60 cases) from the cohort of the University of Illinois at Urbana-Champaign (*Ke et al., 2022*). This cohort contained all faculty, staff, and students at the University of Illinois at Urbana-Champaign, who undergo at least twice weekly quantitative PCR-RT testing during fall of 2020 and spring of 2021.

The viral load data for the Illinois cohort was calculated based on the linear relationship between viral load (copies/mL) and Ct values presented in *Ke et al., 2022*, specific to the measurement method used in this study. Among those 182 cases, we focused only on symptomatic participants. Asymptomatic individuals were excluded since they lack a definable date of symptom onset, which prevents us from establishing a clear baseline for the time axis in parameter estimation. Also, we excluded the participants who had less than three measured viral loads that were not limit detections (i.e. 90 cases from the NFV clinical trial and 54 cases from the University of Illinois were used in this study). Because the trial design discontinued follow-up after two consecutive results below the detection limit, most of these participants already had viral loads near the detection limit at baseline, making reliable model fitting infeasible. The limit of detection for viral load data from the University of Illinois is 1.08 copies/mL. However, the limit of detection for viral load data from NFV clinical trial was unclear. Considering this, we assumed 1.08 copies/ml as the limit of detection for all viral load data. To examine the impact of assumptions regarding the detection limit, we performed a sensitivity analysis. Specifically, for the NFV cohort, we compared the estimated viral dynamics when the detection limit was set to 0 and 2, instead of 1.08. Overall, the results were largely consistent across these scenarios (*Figure 2—figure supplement 5*).

## Viral genome sequencing

The cDNA had been synthesized from RNA of SARS-CoV-2–positive saliva samples. Reverse transcription, multiplex PCR reaction, and Illumina library prep were conducted using a protocol published previously (*Kentaro et al., 2020*). The pooled library was first purified by AMPure XP at 0.8 x concentration and then again at 1.2 x concentration. The purified library was sequenced for 151 cycles at both paired ends in Illumina iSeq100. Sequence analysis was performed using the nf-core/viralcon pipeline (10.5281/zenodo.3901628).

## Quantifying biomarkers in saliva

Total RNA from saliva was extracted with MagMAX CORE Nucleic Acid Purification kits (Applied Biosystems, Foster City, CA). Micro-RNAs were detected using Illumina Hiseq x Ten (Illumina, Inc, San Diego, CA) with data processing by ribodepletion (Genewiz-Azenta, South Plainfield, NJ). To remove technical sequences, the pass filter data in the fastq format were processed by Trimmomatic (v0.30) to be high-quality clean data. Following quality trimming, micro-RNAs were identified and checked using miRDeep2 (*Friedländer et al., 2012*). Normalization of micro-RNA expression among samples and differential expression analysis was carried out using the DESeq2 Bioconductor package.

## Basic clinical data

Basic clinical data including basic demographic characteristics of the study participants, symptoms, and findings of physical examinations and laboratory tests were obtained according to the study protocol (*Hosogaya et al., 2021*). We here used information on age, daily symptoms, results of blood tests, and vital signs of the symptomatic participants in the NFV clinical trial (summarized in *Table 1* and *Supplementary file 1A*).

## Mathematical modeling

To describe SARS-CoV-2 infection dynamics in saliva specimens, we here mainly used the following mathematical model developed in our recent studies (*Kim et al., 2021*; *Jeong et al., 2022*; *Iwanami et al., 2021*):

$$\frac{df(t)}{dt} = -\beta f(t) V(t), \tag{1}$$

$$\frac{dV(t)}{dt} = \gamma f(t) V(t) - \delta V(t). \tag{2}$$

The variables $f(t)$ and $V(t)$ are the fraction of uninfected target cells and the total amount of virus, respectively, and the parameters $\beta$, $\gamma$, and $\delta$ are the rate constant for virus infection, the maximum rate constant for viral replication, and the death rate of infected cells, respectively.

In addition to comparing the simple model (i.e. a target-cell-limited model; *Equations 1-2*), we also used the following 'immune effector cell model' developed in *Ke et al., 2022* for the saliva viral load (see *Figure 2—figure supplement 4*):

$$\frac{dT(t)}{dt} = -\beta T(t) V(t), \tag{3}$$

$$\frac{dE(t)}{dt} = \beta T(t) V(t) - kE(t), \tag{4}$$

$$\frac{dI(t)}{dt} = kE(t) - \delta(t) I(t), \tag{5}$$

$$\frac{dV(t)}{dt} = \pi I(t) - cV(t). \tag{6}$$

The variables $T(t)$, $E(t)$, and $I(t)$ are the total number uninfected target cells, cells in the eclipse phase of infection, and productively infected cells, respectively. The parameters $1/k$, $\pi$, and $c$ are the average duration of the eclipse phase, the virus production rate, and the clearance rate of viruses, respectively. The death rate of infected cells is assumed to be time-dependent to mimic the killing of

infected cells by immune effector cells: $\delta\left(t\right) = \delta_1$ for $t < t_1$ and $\delta\left(t\right) = \delta_1 + \delta_2$ for $t_1 \leq t$. For a detailed explanation of the immune effector cell model, the reader is referred to *Ke et al., 2022*.

## Parameter estimation

A nonlinear mixed-effects modeling approach incorporates fixed effects as well as random effects that describe the inter-participant variability in parameters. Including random effects amounts to a partial pooling of the data of all participants to improve estimates of the parameters applicable across the cases. In this study, viral load data from the Illinois cohort were collected primarily during the early to middle phase of SARS-CoV-2 infection, whereas those from the NFV cohort were obtained mainly during the middle to late phase. In particular, the Illinois cohort provides pre-symptomatic data, while the NFV cohort includes data beyond 14 days after symptom onset, with each cohort lacking the phase represented in the other. Although the NFV cohort lacks pre-symptomatic data, the nonlinear mixed-effects model first derives population-level parameters from all participants and then accounts for individual variability through random effects. This allows pre-symptomatic information from the Illinois cohort to inform the inferred viral dynamics of participants in the NFV cohort. Conversely, for the late phase, the NFV cohort serves the complementary role. By jointly analyzing both cohorts with a nonlinear mixed-effects model to estimate individual-level parameters, we can therefore capture the complete time-course pattern of SARS-CoV-2 infection dynamics across all participants, to the extent possible.

In our analyses, the variable $V\left(t\right)$ in *Equation 2* corresponds to the viral load in saliva for SARS-CoV-2. To fit the patient's viral load data, we used a program MONOLIX 2021R2 (https://www.simulations-plus.com/), implementing maximum likelihood estimation of parameters in nonlinear mixed effect model. The nonlinear mixed-effects model allows a fixed effect as well as interpatient variability. This method estimates each parameter $\theta_i \left(\theta \times e^{\eta_i}\right)$ for each individual where $\theta$ is a fixed effect, and $\eta_i$ is a random effect, and which obeys a Gaussian distribution with mean 0 and standard deviation $\Omega$. Here, we used lognormal distributions as prior distributions of parameters to guarantee the positiveness (i.e. negative values do not biologically make sense). In parameter estimation, as time 0 in the original dataset represents the time of symptom onset, we also estimated the time from infection to symptom onset (corresponding to $\tau$ in *Supplementary file 1B* and *Supplementary file 1C*) along with other parameters. The fixed effect parameters and random effect parameters were estimated by use of the stochastic approximation Expectation/Maximization (SAEM) algorithm and empirical Bayes method, respectively. A right-truncated normal distribution was used in the likelihood function to account for the left censoring of the viral load data (i.e. when the viral load is not detectable; *Samson et al., 2006*). The standard errors were obtained from the Fisher Information Matrix using the linearization method (*Figure 2—figure supplement 3C*). We changed the initial values multiple times to avoid a local minimum of AIC and confirmed the robustness of parameter estimation.

## Unsupervised clustering and stratification of SARS-CoV-2 infection dynamics

Unsupervised random forest clustering was performed on the selected features of the virus infection dynamics, that is, the peak viral load, duration of viral RNA detection, up-slope, and down-slope (rfUtilities package in R). The use of random forest allows us to avoid overfitting by bootstrap aggregating (bagging) and to achieve better generalization performance (*Hastie et al., 2009*). After a random forest dissimilarity (i.e., the distance matrix between all pairs of samples) was obtained, it was visualized with Uniform Manifold Approximation and Projection (UMAP) in a two-dimensional plane and was stratified with spectral clustering (Python scikit-learn). The optimal number of clusters was determined by the eigengap heuristic method.

Furthermore, to evaluate how statistical uncertainty in the estimated viral dynamics might influence the stratification, we propagated parameter uncertainty as follows. For each participant, we resampled the parameters of the mathematical model 100 times within their 95% credible intervals and calculated the corresponding features of reconstructed viral dynamics. We then averaged the features across the 100 replicates to obtain an uncertainty-adjusted feature vector for each participant. Using these vectors, we re-calculated the distance matrix as described above. Then, agreement between the uncertainty-adjusted and original distance matrix was quantified with a permutation-based Mantel test.

## Random forest classifiers for characterizing stratified groups

Random forest classifiers were trained to predict either of the three stratified groups (G1-G3) using randomForest packages in R. The receiver operating characteristic (ROC) curve for each classifier was drawn from out-of-bag (OOB) samples using the pROC package in R. For example, the ROC for G1 is for the random forest classifier predicting 'G1' versus 'G2 or G3'. We list the variables for the supervised random forest in *Table 1*, *Supplementary file 1A*, and *Supplementary file 1B*.

## Statistical analysis

When necessary, the variables were compared among different groups using Fisher's exact test (for categorical variables), analysis of variance (ANOVA, for numerical variables from clinical data with more than two groups), and Kruskal-Wallis ANOVA (for variables from micro-RNA data with more than two groups). Corrections of p-values for multiple testing were performed by the False Discovery Rate (FDR). Also, the variables were investigated for their relationship with features of viral load using the Pearson's correlation coefficient (for variables from clinical data) and the Spearman's correlation coefficient (for variables from micro-RNA data). All statistical analyses were performed using R (version 4.1.3).

## Acknowledgements

This study was supported in part by AMED Research Program 24fk0210154h0001 (to. HP); National Research Foundation of Korea (NRF) grant funded by the Korea government (MSIT) (2022R1C1C2003637) (to KSK); Grant-in-Aid for JSPS Scientific Research (KAKENHI) B 23H03497 (to SI); Grant-in-Aid for Transformative Research Areas 22H05215 (to SI); Grant-in-Aid for Challenging Research (Exploratory) 22K19829 (to SI); AMED CREST 19gm1310002 (to SI); AMED Research Program on Emerging and Re-emerging Infectious Diseases 22fk0108509 (to SI), 23fk0108684 (to SI), 23fk0108685 (to SI); AMED Research Program on HIV/AIDS 22fk0410052 (to SI); AMED Program for Basic and Clinical Research on Hepatitis 22fk0210094 (to SI); AMED Program on the Innovative Development and the Application of New Drugs for Hepatitis B 22fk0310504h0501 (to SI); AMED Strategic Research Program for Brain Sciences 22wm0425011s0302; AMED JP22dm0307009 (to KA); JST MIRAI JPMJMI22G1 (to SI); Moonshot R&D JPMJMS2021 (to KA and SI) and JPMJMS2025 (to SI); JST CREST JPMJCR25Q6 (to SI); the Cabinet Agency for Infectious Disease Crisis Management (to SI); the JIHS Intramural Research Fund (24 rin 002) (to SI); Institute of AI and Beyond at the University of Tokyo (to KA); Shin-Nihon of Advanced Medical Research (to SI); The Japan Prize Foundation (to SI).

## Additional information

### Funding

| Funder | Grant reference number | Author |
|---|---|---|
| Japan Agency for Medical Research and Development | 24fk0210154h0001 | Hyeongki Park |
| National Research Foundation of Korea | 2022R1C1C2003637 | Kwangsu Kim |
| Japan Society for the Promotion of Science | 23H03497 | Shingo Iwami |
| Japan Society for the Promotion of Science | 22H05215 | Shingo Iwami |
| Japan Society for the Promotion of Science | 22K19829 | Shingo Iwami |
| Japan Agency for Medical Research and Development | 19gm1310002 | Shingo Iwami |

| Funder | Grant reference number | Author |
| --- | --- | --- |
| Japan Agency for Medical Research and Development | 22fk0108509 | Shingo Iwami |
| Japan Agency for Medical Research and Development | 23fk0108684 | Shingo Iwami |
| Japan Agency for Medical Research and Development | 23fk0108685 | Shingo Iwami |
| Japan Agency for Medical Research and Development | 22fk0410052 | Shingo Iwami |
| Japan Agency for Medical Research and Development | 22fk0210094 | Shingo Iwami |
| Japan Agency for Medical Research and Development | 22fk0310504h0501 | Shingo Iwami |
| Japan Agency for Medical Research and Development | 22wm0425011s0302 | Kazuyuki Aihara |
| Japan Agency for Medical Research and Development | JP22dm0307009 | Kazuyuki Aihara |
| Japan Science and Technology Agency | JST-Mirai Program JPMJMI22G1 | Shingo Iwami |
| Moonshot Research and Development Program | JPMJMS2021 | Kazuyuki Aihara Shingo Iwami |
| Moonshot Research and Development Program | JPMJMS2025 | Shingo Iwami |
| Japan Science and Technology Agency | JPMJCR25Q6 | Shingo Iwami |
| Cabinet Agency for Infectious Disease Crisis Management | | Shingo Iwami |
| JIHS Intramural Research Fund | 24 rin 002 | Shingo Iwami |
| Institute of AI and Beyond at the University of Tokyo | | Kazuyuki Aihara |
| Shin-Nihon Foundation of Advanced Medical Research | | Shingo Iwami |
| Japan Prize Foundation | | Shingo Iwami |

The funders had no role in study design, data collection and interpretation, or the decision to submit the work for publication.

## Author contributions

Hyeongki Park, Formal analysis, Visualization, Methodology, Writing – original draft, Writing – review and editing; Yoshimura Raiki, Shoya Iwanami, Kwangsu Kim, Keisuke Ejima, Naotoshi Nakamura, Ruian Ke, Formal analysis, Writing – original draft, Writing – review and editing; Kazuyuki Aihara, Formal analysis, Investigation, Writing – original draft, Writing – review and editing; Yoshitsugu Miyazaki, Takashi Umeyama, Takeshi Morita, Koichi Watashi, Christopher B Brooke, Data curation, Investigation, Writing – original draft, Writing – review and editing; Ken Miyazawa, Investigation, Writing – original draft, Writing – review and editing; Shingo Iwami, Conceptualization, Formal analysis, Supervision, Methodology, Writing – original draft, Project administration, Writing – review and editing; Taiga

Miyazaki, Conceptualization, Data curation, Supervision, Funding acquisition, Writing – original draft, Project administration, Writing – review and editing

### Author ORCIDs
Hyeongki Park (ID) http://orcid.org/0009-0002-6507-0731
Naotoshi Nakamura (ID) https://orcid.org/0000-0002-4798-3660
Kazuyuki Aihara (ID) https://orcid.org/0000-0002-4602-9816
Christopher B Brooke (ID) https://orcid.org/0000-0002-6815-1193
Shingo Iwami (ID) https://orcid.org/0000-0002-1780-350X

### Ethics
The NFV clinical trial was approved by the institutional review board of Nagasaki University Hospital (approval number: I20-001) and is registered with the Japan Registry of Clinical Trials (jRCT2071200023). All participants provided written, informed consent for secondary use of clinical information and samples. The present study was approved by the ethics committee of Nagoya University (approval number: hc 22-01).

Reviewer #1 (Public Review): https://doi.org/10.7554/eLife.96032.3.sa1
Reviewer #2 (Public Review): https://doi.org/10.7554/eLife.96032.3.sa2
Reviewer #3 (Public Review): https://doi.org/10.7554/eLife.96032.3.sa3
Author response https://doi.org/10.7554/eLife.96032.3.sa4

## Additional files

### Supplementary files
Supplementary file 1. Supplementary tables.

MDAR checklist

### Data availability
The data generated and analyzed in this study include individual-level clinical, microRNA, and viral load data. All data are managed in accordance with the ethical standards and data management policies of the affiliated institutions. Due to the inclusion of detailed longitudinal and regional information, even de-identified versions of these datasets carry a non-negligible risk of participant re-identification, and the informed consent forms and institutional review board (IRB) approvals at the participating institutions do not permit public release of individual-level data. For these ethical and legal reasons, neither the raw nor de-identified clinical, microRNA, and viral load datasets can be publicly shared. However, the reconstructed viral load dynamics derived from the mathematical models described in this study are fully reproducible using the publicly available analysis code, and all scripts required to reproduce the model-based viral load dynamics and figures are available on Zenodo (https://doi.org/10.5281/zenodo.18125739). Researchers seeking access to the restricted individual-level datasets may contact the corresponding author, Taiga Miyazaki (taiga_miyazaki@med.miyazaki-u.ac.jp); access may be granted upon reasonable request and subject to approval by the relevant institutional review boards of all participating institutions.

The following dataset was generated:

| Author(s) | Year | Dataset title | Dataset URL | Database and Identifier |
|---|---|---|---|---|
| Park H | 2026 | Stratification of viral shedding patterns in saliva of COVID-19 patients | https://doi.org/10.5281/zenodo.18125739 | Zenodo, 10.5281/zenodo.18125739 |

The following previously published dataset was used:

| Author(s) | Year | Dataset title | Dataset URL | Database and Identifier |
|---|---|---|---|---|
| Ke R, Martinez PP, Smith RL, Gibson LL, Mirza A, Conte M, Gallagher N, Luo CH, Jarrett J, Zhou R, Conte A, Liu T, Farjo M, Walden KKO, Rendon G, Fields CJ, Wang L, Fredrickson R, Edmonson DC, Baughman ME, Chiu KK, Choi H, Scardina KR, Bradley S, Gloss SL, Reinhart C, Yedetore J, Quicksall J, Owens AN, Broach J, Barton B, Lazar P, Heetderks WJ, Robinson ML, Mostafa HH, Manabe YC, Pekosz A, McManus DD, Brooke CB | 2022 | Daily sampling of early SARS-CoV-2 infection reveals substantial heterogeneity in infectiousness | https://www.ncbi.nlm.nih.gov/bioproject/?term=PRJNA809434 | NCBI BioProject, PRJNA809434 |

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
