## [Editor Report · eLife Assessment]

This **important** work attempts to understand observed variability in oral shedding of SARS-CoV-2 and suggests that routine clinical factors are not determinative. The evidence supporting the conclusion is **solid** though the limited clinical heterogeneity of the included cohorts, the lack of COVID vaccination, and the absence of comprehensive viral load data for model training, makes the results difficult to generalize to contemporaneous COVID-19 conditions. This study may be of interest to virologists, public health officials and clinicians.

---

## [Referee Report · Reviewer #1 (Public Review)]

Summary:

This study by Park and colleagues uses longitudinal saliva viral load data from two cohorts (one in the US and one in Japan from a clinical trial) in the pre-vaccine era to subset viral shedding kinetics and then use machine learning to attempt to identify clinical correlates of different shedding patterns. The stratification method identifies three separate shedding patterns discriminated by peak viral load, shedding duration, and clearance slope. The authors also assess micro-RNAs as potential biomarkers of severity but do not identify any clear relationships with viral kinetics.

Strengths:

The cohorts are well developed, the mathematical model appears to capture shedding kinetics fairly well, the clustering seems generally appropriate, and the machine learning analysis is a sensible, albeit exploratory approach. The micro-RNA analysis is interesting and novel.

---

## [Referee Report · Reviewer #2 (Public Review)]

Summary:

This study argues it has found that it has stratified viral kinetics for saliva specimens into three groups by the duration of "viral shedding"; the authors could not identify clinical data or microRNAs that correlate with these three groups.

Strengths:

The question of whether there is a stratification of viral kinetics is interesting.

---

## [Referee Report · Reviewer #3 (Public Review)]

The article presents a comprehensive study on the stratification of viral shedding patterns in saliva among COVID-19 patients. The authors analyze longitudinal viral load data from 144 mildly symptomatic patients using a mathematical model, identifying three distinct groups based on the duration of viral shedding. Despite analyzing a wide range of clinical data and micro-RNA expression levels, the study could not find significant predictors for the stratified shedding patterns, highlighting the complexity of SARS-CoV-2 dynamics in saliva. The research underscores the need for identifying biomarkers to improve public health interventions and acknowledges several limitations, including the lack of consideration of recent variants, the sparsity of information before symptom onset, and the focus on symptomatic infections.

The manuscript is well-written, with the potential for enhanced clarity in explaining statistical methodologies. This work could inform public health strategies and diagnostic testing approaches.

Comments on the revised version from the editor:

The authors comprehensively addressed the concerns of all 3 reviewers. We are thankful for their considerable efforts to do so. Certain limitations remain unavoidable such as the lack of immunologic diversity among included study participants and lack of contemporaneous variants of concern.

One remaining issue is the continued use of the target cell limited model which is sufficient in most cases, but misses key datapoints in certain participants. In particular, viral rebound is poorly described by this model. Even if viral rebound does not place these cases in a unique cluster, it is well understood that viral rebound is of clinical significance.

In addition, the use of microRNAs as a potential biomarker is still not fully justified. In other words, are there specific microRNAs that have a pre-existing mechanistic basis for relating to higher or lower viral loads? As written it still feels like microRNA was included in the analysis simply because the data existed.

---

## [Author Response]

The following is the authors’ response to the original reviews.

**Reviewer #1 (Public review)**
Summary:This study by Park and colleagues uses longitudinal saliva viral load data from two cohorts (one in the US and one in Japan from a clinical trial) in the pre-vaccine era to subset viral shedding kinetics and then use machine learning to attempt to identify clinical correlates of different shedding patterns. The stratification method identifies three separate shedding patterns discriminated by peak viral load, shedding duration, and clearance slope. The authors also assess micro-RNAs as potential biomarkers of severity but do not identify any clear relationships with viral kinetics.Strengths:The cohorts are well developed, the mathematical model appears to capture shedding kinetics fairly well, the clustering seems generally appropriate, and the machine learning analysis is a sensible, albeit exploratory approach. The micro-RNA analysis is interesting and novel.Weaknesses:The conclusions of the paper are somewhat supported by the data but there are certain limitations that are notable and make the study's findings of only limited relevance to current COVID-19 epidemiology and clinical conditions.

We sincerely appreciate the reviewer’s thoughtful and constructive comments, which have been invaluable in improving the quality of our study. We have carefully revised the manuscript to address all points raised.

(1) The study only included previously uninfected, unvaccinated individuals without the omicron variant. It has been well documented that vaccination and prior infection both predict shorter duration shedding. Therefore, the study results are no longer relevant to current COVID-19 conditions. This is not at all the authors' fault but rather a difficult reality of much retrospective COVID research.

Thank you for your comment. We agree with the review’s comment that some of our results could not provide insight into the current COVID-19 conditions since most people have either already been infected with COVID-19 or have been vaccinated. We revised our manuscript to discuss this (page 22, lines 364-368). Nevertheless, we believe it is novel that we have extensively investigated the relationship between viral shedding patterns in saliva and a wide range of clinical and microRNA data, and that developing a method to do so remains important. This is important for providing insight into early responses to novel emerging viral diseases in the future. Therefore, we still believe that our findings are valuable.

(2) The target cell model, which appears to fit the data fairly well, has clear mechanistic limitations. Specifically, if such a high proportion of cells were to get infected, then the disease would be extremely severe in all cases. The authors could specify that this model was selected for ease of use and to allow clustering, rather than to provide mechanistic insight. It would be useful to list the AIC scores of this model when compared to the model by Ke.

Thank you for your feedback and suggestion regarding our mathematical model. As the reviewer pointed out, in this study, we adopted a simple model (target cell-limited model) to focus on reconstruction of viral dynamics and stratification of shedding patterns rather than exploring the mechanism of viral infection in detail. Nevertheless, we believe that the target cell-limited model provides reasonable reconstructed viral dynamics as it has been used in many previous studies. We revised manuscript to clarify this point (page 10, lines 139-144). Also, we revised our manuscript to provide more detailed description of the model comparison along with information about AIC (page 10, lines 130-135).

(3) Line 104: I don't follow why including both datasets would allow one model to work better than the other. This requires more explanation. I am also not convinced that non-linear mixed effects approaches can really be used to infer early model kinetics in individuals from one cohort by using late viral load kinetics in another (and vice versa). The approach seems better for making populationlevel estimates when there is such a high amount of missing data.

Thank you for your feedback. We recognized that our explanation was insufficient by your comment. We intended to describe that, rather than comparing performance of the two models, data fitting can be performed with same level for both models by including both datasets. We revised the manuscript to clarify this point (page 10, lines 135-139).

Additionally, we agree that nonlinear mixed effects models are a useful approach for performing population-level estimates of missing data. On the other hand, in addition, the nonlinear mixed effects model has the advantage of making the reasonable parameter estimation for each individual with not enough data points by considering the distribution of parameters of other individuals. Paying attention to these advantages, we adopted a nonlinear mixed effects model in our study. We also revised the manuscript to clarify this (page 27, lines 472-483).

(4) Along these lines, the three clusters appear to show uniform expansion slopes whereas the NBA cohort, a much larger cohort that captured early and late viral loads in most individuals, shows substantial variability in viral expansion slopes. In Figure 2D: the upslope seems extraordinarily rapid relative to other cohorts. I calculate a viral doubling time of roughly 1.5 hours. It would be helpful to understand how reliable of an estimate this is and also how much variability was observed among individuals.

We appreciate your detailed feedback on the estimated up-slope of viral dynamics. As the reviewer noted, the pattern differs from that observed in the NBA cohort, which may be due to their measurement of viral load from upper respiratory tract swabs. In our estimation, the mean and standard deviation of the doubling time (defined as ln2/(𝛽𝑇_0_𝑝𝑐^−1^ − 𝛿)) were 1.44 hours and 0.49 hours, respectively. Although direct validation of these values is challenging, several previous studies, including our own, have reported that viral loads in saliva increase more rapidly than in the upper respiratory tract swabs, reaching their peak sooner. Thus, we believe that our findings are consistent with those of previous studies. We revised our manuscript to discuss this point with additional references (page 20, lines 303-311).

(5) A key issue is that a lack of heterogeneity in the cohort may be driving a lack of differences between the groups. Table 1 shows that Sp02 values and lab values that all look normal. All infections were mild. This may make identifying biomarkers quite challenging.

Thank you for your comment regarding heterogeneity in the cohort. Although the NFV cohort was designed for COVID-19 patients who were either mild or asymptomatic, we have addressed this point and revised the manuscript to discuss it (page 21, lines 334-337).

(6) Figure 3A: many of the clinical variables such as basophil count, Cl, and protein have very low pre-test probability of correlating with virologic outcome.

Thank you for your comment regarding some clinical information we used in our study. We revised our manuscript to discuss this point (page 21, lines 337-338).

(7) A key omission appears to be micoRNA from pre and early-infection time points. It would be helpful to understand whether microRNA levels at least differed between the two collection timepoints and whether certain microRNAs are dynamic during infection.

Thank you for your comment regarding the collection of micro-RNA data. As suggested by the reviewer, we compared micro-RNA levels between two time points using pairwise t-tests and Mann-Whitney U tests with FDR correction. As a result, no micro-RNA showed a statistically significant difference. This suggests that micro-RNA levels remain relatively stable during the course of infection, at least for mild or asymptomatic infection, and may therefore serve as a biomarker independent of sampling time. We have revised the manuscript to include this information (page 17, lines 259-262).

(8) The discussion could use a more thorough description of how viral kinetics differ in saliva versus nasal swabs and how this work complements other modeling studies in the field.

We appreciate the reviewer’s thoughtful feedback. As suggested, we have added a discussion comparing our findings with studies that analyzed viral dynamics using nasal swabs, thereby highlighting the differences between viral dynamics in saliva and in the upper respiratory tract. To ensure a fair and rigorous comparison, we referred to studies that employed the same mathematical model (i.e., Eqs.(1-2)). Accordingly, we revised the manuscript and included additional references (page 20, lines 303-311).

Furthermore, we clarified the significance of our study in two key aspects. First, it provides a detailed analysis of viral dynamics in saliva, reinforcing our previous findings from a single cohort by extending them across multiple cohorts. Second, this study uniquely examines whether viral dynamics in saliva can be directly predicted by exploring diverse clinical data and micro-RNAs. Notably, cohorts that have simultaneously collected and reported both viral load and a broad spectrum of clinical data from the same individuals, as in our study, are exceedingly rare. We revised the manuscript to clarify this point (page 20, lines 302-311).

(9) The most predictive potential variables of shedding heterogeneity which pertain to the innate and adaptive immune responses (virus-specific antibody and T cell levels) are not measured or modeled.

Thank you for your comment. We agree that antibody and T cell related markers may serve as the most powerful predictors, as supported by our own study [S. Miyamoto et al., PNAS (2023), ref. 24] as well as previous reports. While this point was already discussed in the manuscript, we have revised the text to make it more explicit (page 21, lines 327-328).

(10) I am curious whether the models infer different peak viral loads, duration, expansion, and clearance slopes between the 2 cohorts based on fitting to different infection stage data.

Thank you for your comment. We compared features between 2 cohorts as reviewer suggested. As a result, a statistically significant difference between the two cohorts (i.e., p-value ≤ 0.05 from the t-test) was observed only at the peak viral load, with overall trends being largely similar. At the peak, the mean value was 7.5 log_10_ (copies/mL) in the Japan cohort and 8.1 log_10_ (copies/mL) in the Illinois cohort, with variances of 0.88 and 0.87, respectively, indicating comparable variability.

**Reviewer #2 (Public review)**
Summary:This study argues it has found that it has stratified viral kinetics for saliva specimens into three groups by the duration of "viral shedding"; the authors could not identify clinical data or microRNAs that correlate with these three groups.Strengths:The question of whether there is a stratification of viral kinetics is interesting.Weaknesses:The data underlying this work are not treated rigorously. The work in this manuscript is based on PCR data from two studies, with most of the data coming from a trial of nelfinavir (NFV) that showed no effect on the duration of SARS-CoV-2 PCR positivity. This study had no PCR data before symptom onset, and thus exclusively evaluated viral kinetics at or after peak viral loads. The second study is from the University of Illinois; this data set had sampling prior to infection, so has some ability to report the rate of "upswing." Problems in the analysis here include:

We are grateful to the reviewer for the constructive feedback, which has greatly enhanced the quality of our study. In response, we have carefully revised the manuscript to address all comments.

The PCR Ct data from each study is treated as equivalent and referred to as viral load, without any reports of calibration of platforms or across platforms. Can the authors provide calibration data and justify the direct comparison as well as the use of "viral load" rather than "Ct value"? Can the authors also explain on what basis they treat Ct values in the two studies as identical?

Thank you for your comment regarding description of viral load data. We recognized the lack of explanation for the integration of viral load data by reviewer's comment. We calculated viral load from Ct value using linear regression equations between Ct and viral load for each study's measurement method, respectively. We revised the manuscript to clarify this point in the section of Saliva viral load data in Methods.

The limit of detection for the NFV PCR data was unclear, so the authors assumed it was the same as the University of Illinois study. This seems a big assumption, as PCR platforms can differ substantially. Could the authors do sensitivity analyses around this assumption?

Thank you for your comment regarding the detection limit for viral load data. As reviewer suggested, we conducted sensitivity analysis for assumption of detection limit for the NFV dataset. Specifically, we performed data fitting in the same manner for two scenarios: when the detection limit of NFV PCR was lower (0 log_10_ copies/mL) or higher (2 log_10_ copies/mL) than that of the Illinois data (1.08 log_10_ copies/mL), and compared the results.

As a result, we obtained largely comparable viral dynamics in most cases (Supplementary Fig 6). When comparing the AIC values, we observed that the AIC for the same censoring threshold was 6836, whereas it increased to 7403 under the low censoring threshold and decreased to 6353 under the higher censoring threshold. However, this difference may be attributable to the varying number of data points treated as below the detection limit. Specifically, when the threshold is set higher, more data are treated as below the detection limit, which may result in a more favorable error calculation. To discuss this point, we have added a new figure (Supplementary Fig 6) and revised the manuscript accordingly (page 25, lines 415-418).

The authors refer to PCR positivity as viral shedding, but it is viral RNA detection (very different from shedding live/culturable virus, as shown in the Ke et al. paper). I suggest updating the language throughout the manuscript to be precise on this point.

We appreciate the reviewer’s feedback regarding the terminology used for viral shedding. In response, we have revised all instances of “viral shedding” to “viral RNA detection” throughout the manuscript as suggested.

Eyeballing extended data in Figure 1, a number of the putative long-duration infections appear to be likely cases of viral RNA rebound (for examples, see S01-16 and S01-27). What happens if all the samples that look like rebound are reanalyzed to exclude the late PCR detectable time points that appear after negative PCRs?

We sincerely thank the reviewer for the valuable suggestion. In response, we established a criterion to remove data that appeared to exhibit rebound and subsequently performed data fitting

(see Author response image 1 below). The criterion was defined as: “any data that increase again after reaching the detection limit in two measurements are considered rebound and removed.” As a result, 15 out of 144 cases were excluded due to insufficient usable data, leaving 129 cases for analysis. Using a single detection limit as the criterion would have excluded too many data points, while defining the criterion solely based on the magnitude of increase made it difficult to establish an appropriate “threshold for increase.”

The fitting result indicates that the removal of rebound data may influence the fitting results; however, direct comparison of subsequent analyses, such as clustering, is challenging due to the reduced sample size. Moreover, the results can vary substantially depending on the criterion used to define rebound, and establishing a consistent standard remains difficult. Accordingly, we retained the current analysis and have added a discussion of rebound phenomena in the Discussion section as a limitation (page 22, lines 355-359). We once again sincerely appreciate the reviewer’s insightful and constructive suggestion.

**Author response image 1. sa4fig1:** Comparison of model fits before and after removing data suspected of rebound. Black dots represent observed measurements, and the black and yellow curves show the fitted viral dynamics for the full dataset and the dataset with rebound data removed, respectively.

There's no report of uncertainty in the model fits. Given the paucity of data for the upslope, there must be large uncertainty in the up-slope and likely in the peak, too, for the NFV data. This uncertainty is ignored in the subsequent analyses. This calls into question the efforts to stratify by the components of the viral kinetics. Could the authors please include analyses of uncertainty in their model fits and propagate this uncertainty through their analyses?

We sincerely appreciate the reviewer’s detailed feedback on model uncertainty. To address this point, we revised Extended Fig 1 (now renumbered as Supplementary Fig 1) to include 95% credible intervals computed using a bootstrap approach. In addition, to examine the potential impact of model uncertainty on stratified analyses, we reconstructed the distance matrix underlying stratification by incorporating feature uncertainty. Specifically, for each individual, we sampled viral dynamics within the credible interval and averaged the resulting feature, and build the distance matrix using it. We then compared this uncertainty-adjusted matrix with the original one using the Mantel test, which showed a strong correlation (r = 0.72, p < 0.001). Given this result, we did not replace the current stratification but revised the manuscript to provide this information through Result and Methods sections (page 11, lines 159-162 and page 28, lines 512-519). Once again, we are deeply grateful for this insightful comment.

The clinical data are reported as a mean across the course of an infection; presumably vital signs and blood test results vary substantially, too, over this duration, so taking a mean without considering the timing of the tests or the dynamics of their results is perplexing. I'm not sure what to recommend here, as the timing and variation in the acquisition of these clinical data are not clear, and I do not have a strong understanding of the basis for the hypothesis the authors are testing.

We appreciate the reviewers' feedback on the clinical data. We recognized that the manuscript lacked description of the handling of clinical data by your comment. In this research, we focused on finding “early predictors” which could provide insight into viral shedding patterns. Thus, we used clinical data measured in the earliest time (date of admission) for each patient. Another reason is that the date of admission is the almost only time point at which complete clinical data without any missing values are available for all participants. We revised our manuscript to clarify this point (page 5, lines 90-95).

It's unclear why microRNAs matter. It would be helpful if the authors could provide more support for their claims that (1) microRNAs play such a substantial role in determining the kinetics of other viruses and (2) they play such an important role in modulating COVID-19 that it's worth exploring the impact of microRNAs on SARS-CoV-2 kinetics. A link to a single review paper seems insufficient justification. What strong experimental evidence is there to support this line of research?

We appreciate the reviewer’s comments regarding microRNA. Based on this feedback, we recognized the need to clarify our rationale for selecting microRNAs as the analyte. The primary reason was that our available specimens were saliva, and microRNAs are among the biomarkers that can be reliably measured in saliva. At the same time, previous studies have reported associations between microRNAs and various diseases, which led us to consider the potential relevance of microRNAs to viral dynamics, beyond their role as general health indicators. To better reflect this context, we have added supporting references (page 17, lines 240-243).

**Reviewer #3 (Public review)**
The article presents a comprehensive study on the stratification of viral shedding patterns in saliva among COVID-19 patients. The authors analyze longitudinal viral load data from 144 mildly symptomatic patients using a mathematical model, identifying three distinct groups based on the duration of viral shedding. Despite analyzing a wide range of clinical data and micro-RNA expression levels, the study could not find significant predictors for the stratified shedding patterns, highlighting the complexity of SARS-CoV-2 dynamics in saliva. The research underscores the need for identifying biomarkers to improve public health interventions and acknowledges several limitations, including the lack of consideration of recent variants, the sparsity of information before symptom onset, and the focus on symptomatic infections.The manuscript is well-written, with the potential for enhanced clarity in explaining statistical methodologies. This work could inform public health strategies and diagnostic testing approaches. However, there is a thorough development of new statistical analysis needed, with major revisions to address the following points:

We sincerely appreciate the thoughtful feedback provided by Reviewer #3, particularly regarding our methodology. In response, we conducted additional analyses and revised the manuscript accordingly. Below, we address the reviewer’s comments point by point.

(1) Patient characterization & selection: Patient immunological status at inclusion (and if it was accessible at the time of infection) may be the strongest predictor for viral shedding in saliva. The authors state that the patients were not previously infected by SARS-COV-2. Was Anti-N antibody testing performed? Were other humoral measurements performed or did everything rely on declaration? From Figure 1A, I do not understand the rationale for excluding asymptomatic patients. Moreover, the mechanistic model can handle patients with only three observations, why are they not included? Finally, the 54 patients without clinical data can be used for the viral dynamics fitting and then discarded for the descriptive analysis. Excluding them can create a bias. All the discarded patients can help the virus dynamics analysis as it is a population approach. Please clarify. In Table 1 the absence of sex covariate is surprising.

We appreciate the detailed feedback from the reviewer regarding patient selection. We relied on the patient's self-declaration to determine the patient's history of COVID-19 infection and revised the manuscript to specify this (page 6, lines 83-84).

In parameter estimation, we used the date of symptom onset for each patient so that we establish a baseline of the time axis as clearly as possible, as we did in our previous works. Accordingly, asymptomatic patients who do not have information on the date of symptom onset were excluded from the analysis. Additionally, in the cohort we analyzed, for patients excluded due to limited number of observations (i.e., less than 3 points), most patients already had a viral load close to the detection limit at the time of the first measurement. This is due to the design of clinical trial, as if a negative result was obtained twice in a row, no further follow-up sampling was performed. These patients were excluded from the analysis because it hard to get reasonable fitting results. Also, we used 54 patients for the viral dynamics fitting and then only used the NFV cohort for clinical data analysis. We acknowledge that our description may have confused readers. We revised our manuscript to clarify these points regarding patient selecting for data fitting (page 6, lines 96-102, page 24, lines 406-407, and page 7, lines 410-412). In addition, we realized, thanks to the reviewer’s comment, that gender information was missing in Table 1. We appreciate this observation and have revised the table to include gender (we used gender in our analysis).

(2) Exact study timeline for explanatory covariates: I understand the idea of finding « early predictors » of long-lasting viral shedding. I believe it is key and a great question. However, some samples (Figure 4A) seem to be taken at the end of the viral shedding. I am not sure it is really easier to micro-RNA saliva samples than a PCR. So I need to be better convinced of the impact of the possible findings. Generally, the timeline of explanatory covariate is not described in a satisfactory manner in the actual manuscript. Also, the evaluation and inclusion of the daily symptoms in the analysis are unclear to me.

We appreciate the reviewer’s feedback regarding the collection of explanatory variables. As noted, of the two microRNA samples collected from each patient, one was obtained near the end of viral shedding. This was intended to examine potential differences in microRNA levels between the early and late phases of infection. No significant differences were observed between the two time points, and using microRNA from either phase alone or both together did not substantially affect predictive accuracy for stratified groups. Furthermore, microRNA collection was motivated primarily by the expectation that it would be more sensitive to immune responses, rather than by ease of sampling. We have revised the manuscript to clarify these points regarding microRNA (page 17, lines 243-245 and 259-262).

Furthermore, as suggested by the reviewer, we have also strengthened the explanation regarding the collection schedule of clinical information and the use of daily symptoms in the analysis (page 6, lines 90-95, page 14, lines 218-220,).

(3) Early Trajectory Differentiation: The model struggles to differentiate between patients' viral load trajectories in the early phase, with overlapping slopes and indistinguishable viral load peaks observed in Figures 2B, 2C, and 2D. The question arises whether this issue stems from the data, the nature of Covid-19, or the model itself. The authors discuss the scarcity of pre-symptom data, primarily relying on Illinois patients who underwent testing before symptom onset. This contrasts earlier statements on pages 5-6 & 23, where they claim the data captures the full infection dynamics, suggesting sufficient early data for pre-symptom kinetics estimation. The authors need to provide detailed information on the number or timing of patient sample collections during each period.

Thank you for the reviewer’s thoughtful comments. The model used in this study [Eqs.(1-2)] has been employed in numerous prior studies and has successfully identified viral dynamics at the individual level. In this context, we interpret the rapid viral increase observed across participants as attributable to characteristics of SARS-CoV-2 in saliva, an interpretation that has also been reported by multiple previous studies. We have added the relevant references and strengthened the corresponding discussion in the manuscript (page 20, lines 303-311).

We acknowledge that our explanation of how the complementary relationship between the two cohorts contributes to capturing infection dynamics was not sufficiently clear. As described in the manuscript, the Illinois cohort provides pre-symptomatic data, whereas the NFV cohort offers abundant end-phase data, thereby compensating for each other’s missing phases. By jointly analyzing the two cohorts with a nonlinear mixed-effects model, we estimated viral dynamics at the individual-level. This approach first estimates population-level parameters (fixed effects) using data from all participants and then incorporates random effects to account for individual variability, yielding the most plausible parameter values.

Thus, even when early-phase data are lacking in the NFV cohort, information from the Illinois cohort allows us to infer most reasonable dynamics, and the reverse holds true for the end phase. In this context, we argued that combining the two cohorts enables mathematical modeling to capture infection dynamics at the individual level. Recognizing that our earlier description could be misleading, we have carefully reinforced the relevant description (page 27, lines 472-483). In addition, as suggested by the reviewer, we have added information on the number of data samples available for each phase in both cohorts (page 7, lines 106-109).

(4) Conditioning on the future: Conditioning on the future in statistics refers to the problematic situation where an analysis inadvertently relies on information that would not have been available at the time decisions were made or data were collected. This seems to be the case when the authors create micro-RNA data (Figure 4A). First, when the sampling times are is something that needs to be clarified by the authors (for clinical outcomes as well). Second, proper causal inference relies on the assumption that the cause precedes the effect. This conditioning on the future may result in overestimating the model's accuracy. This happens because the model has been exposed to the outcome it's supposed to predict. This could question the - already weak - relation with mir-1846 level.

We appreciate the reviewer’s detailed feedback. As noted in Reply to Comments 2, we collected micro-RNA samples at two time points, near the peak of infection dynamics and at the end stage, and found no significant differences between them. This suggests that micro-RNA levels are not substantially affected by sampling time. Indeed, analyses conducted using samples from the peak, late stage, or both yielded nearly identical results in relation to infection dynamics. To clarify this point, we revised the manuscript by integrating this explanation with our response in Reply to Comments 2 (page 17, lines 259-262). In addition, now we also revised manuscript to clarify sampling times of clinical information and micro-RNA (page 6, lines 90-95).

(5) Mathematical Model Choice Justification and Performance: The paper lacks mention of the practical identifiability of the model (especially for tau regarding the lack of early data information). Moreover, it is expected that the immune effector model will be more useful at the beginning of the infection (for which data are the more parsimonious). Please provide AIC for comparison, saying that they have "equal performance" is not enough. Can you provide at least in a point-by-point response the VPC & convergence assessments?

We appreciate the reviewer’s detailed feedback regarding the mathematical model. We acknowledge the potential concern regarding the practical identifiability of tau (incubation period), particularly given the limited early-phase data. In our analysis, however, the nonlinear mixed-effects model yielded a population-level estimate of 4.13 days, which is similar with previously reported incubation periods for COVID-19. This concordance suggests that our estimate of tau is reasonable despite the scarcity of early data.

For model comparison, first, we have added information on the AIC of the two models to the manuscript as suggested by the reviewer (page 10, lines 130-135). One point we would like to emphasize is that we adopted a simple target cell-limited model in this study, aiming to focus on reconstruction of viral dynamics and stratification of shedding patterns rather than exploring the mechanism of viral infection in detail. Nevertheless, we believe that the target cell-limited model provides reasonable reconstructed viral dynamics as it has been used in many previous studies. We revised manuscript to clarify this (page 10, lines 135-144).

Furthermore, as suggested, we have added the VPC and convergence assessment results for both models, together with explanatory text, to the manuscript (Supplementary Fig 2, Supplementary Fig 3, and page 10, lines 130-135). In the VPC, the observed 5th, 50th, and 95th percentiles were generally within the corresponding simulated prediction intervals across most time points. Although minor deviations were noted in certain intervals, the overall distribution of the observed data was well captured by the models, supporting their predictive performance (Supplementary Fig 2). In addition, the log-likelihood and SAEM parameter trajectories stabilized after the burn-in phase, confirming appropriate convergence (Supplementary Fig 3).

(6) Selected features of viral shedding: I wonder to what extent the viral shedding area under the curve (AUC) and normalized AUC should be added as selected features.

We sincerely appreciate the reviewer’s valuable suggestion regarding the inclusion of additional features. Following this recommendation, we considered AUC (or normalized AUC) as an additional feature when constructing the distance matrix used for stratification. We then evaluated the similarity between the resulting distance matrix and the original one using the Mantel test, which showed a very high correlation (r = 0.92, p < 0.001). This indicates that incorporating AUC as an additional feature does not substantially alter the distance matrix. Accordingly, we have decided to retain the current stratification analysis, and we sincerely thank the reviewer once again for this interesting suggestion.

(7) Two-step nature of the analysis: First you fit a mechanistic model, then you use the predictions of this model to perform clustering and prediction of groups (unsupervised then supervised). Thus you do not propagate the uncertainty intrinsic to your first estimation through the second step, ie. all the viral load selected features actually have a confidence bound which is ignored. Did you consider a one-step analysis in which your covariates of interest play a direct role in the parameters of the mechanistic model as covariates? To pursue this type of analysis SCM (Johnson et al. Pharm. Res. 1998), COSSAC (Ayral et al. 2021 CPT PsP), or SAMBA (Prague et al. CPT PsP 2021) methods can be used. Did you consider sampling on the posterior distribution rather than using EBE to avoid shrinkage?

Thank you for the reviewer’s detailed suggestions regarding our analysis. We agree that the current approach does not adequately account for the impact of uncertainty in viral dynamics on the stratified analyses. As a first step, we have revised Extended Data Fig 1 (now renumbered as Supplementary Fig 1) to include 95% credible intervals computed using a bootstrap approach, to present the model-fitting uncertainty more explicitly. Then, to examine the potential impact of model uncertainty on stratified analyses, we reconstructed the distance matrix underlying stratification by incorporating feature uncertainty. Specifically, for each individual, we sampled viral dynamics within the credible interval and averaged the resulting feature, and build the distance matrix using it. We then compared this uncertainty-adjusted matrix with the original one using the Mantel test, which showed a strong correlation (r = 0.72, p < 0.001). Given this result, we did not replace the current stratification but revised the manuscript to provide this information (page 11, lines 159-162 and page 28, 512-519).

Furthermore, we carefully considered the reviewer’s proposed one-step analysis. However, implementation was constrained by data-fitting limitations. Concretely, clinical information is available only in the NFV cohort. Thus, if these variables are to be entered directly as covariates on the parameters, the Illinois cohort cannot be included in the data-fitting process. Yet the NFV cohort lacks any pre-symptomatic observations, so fitting the model to that cohort alone does not permit a reasonable (well-identified/robust) fitting result. While we were unable to implement the suggestion under the current data constraints, we sincerely appreciate the reviewer’s thoughtful and stimulating proposal.

(8) Need for advanced statistical methods: The analysis is characterized by a lack of power. This can indeed come from the sample size that is characterized by the number of data available in the study. However, I believe the power could be increased using more advanced statistical methods. At least it is worth a try. First considering the unsupervised clustering, summarizing the viral shedding trajectories with features collapses longitudinal information. I wonder if the R package « LongituRF » (and associated method) could help, see Capitaine et al. 2020 SMMR. Another interesting tool to investigate could be latent class models R package « lcmm » (and associated method), see ProustLima et al. 2017 J. Stat. Softwares. But the latter may be more far-reached.

Thank you for the reviewer’s thoughtful suggestions regarding our unsupervised clustering approach. The R package “LongitiRF” is designed for supervised analysis, requiring a target outcome to guide the calculation of distances between individuals (i.e., between viral dynamics). In our study, however, the goal was purely unsupervised clustering, without any outcome variable, making direct application of “LongitiRF” challenging.

Our current approach (summarizing each dynamic into several interpretable features and then using Random Forest proximities) allows us to construct a distance matrix in an unsupervised manner. Here, the Random Forest is applied in “proximity mode,” focusing on how often dynamics are grouped together in the trees, independent of any target variable. This provides a practical and principled way to capture overall patterns of dynamics while keeping the analysis fully unsupervised.

Regarding the suggestion to use latent class mixed models (R package “lcmm”), we also considered this approach. In our dataset, each subject has dense longitudinal measurements, and at many time points, trajectories are very similar across subjects, resulting in minimal inter-individual differences. Consequently, fitting multi-class latent class mixed models (ng ≥ 2) with random effects or mixture terms is numerically unstable, often producing errors such as non-positive definite covariance matrices or failure to generate valid initial values. Although one could consider using only the time points with the largest differences, this effectively reduces the analysis to a feature-based summary of dynamics. Such an approach closely resembles our current method and contradicts the goal of clustering based on full longitudinal information.

Taken together, although we acknowledge that incorporating more longitudinal information is important, we believe that our current approach provides a practical, stable, and informative solution for capturing heterogeneity in viral dynamics. We would like to once again express our sincere gratitude to the reviewer for this insightful suggestion.

(9) Study intrinsic limitation: All the results cannot be extended to asymptomatic patients and patients infected with recent VOCs. It definitively limits the impact of results and their applicability to public health. However, for me, the novelty of the data analysis techniques used should also be taken into consideration.

We appreciate your positive evaluation of our research approach and acknowledge that, as noted in the Discussion section as our first limitation, our analysis may not provide valid insights into recent VOCs or all populations, including asymptomatic individuals. Nonetheless, we believe it is novel that we extensively investigated the relationship between viral shedding patterns in saliva and a wide range of clinical and micro-RNA data. Our findings contribute to a deeper and more quantitative understanding of heterogeneity in viral dynamics, particularly in saliva samples. To discuss this point, we revised our manuscript (page 22, lines 364-368).

Strengths are:Unique data and comprehensive analysis.Novel results on viral shedding.Weaknesses are:Limitation of study design.The need for advanced statistical methodology.
**Reviewer #1 (Recommendations For The Authors):**
Line 8: In the abstract, it would be helpful to state how stratification occurred.

We thank the reviewer for the feedback, and have revised the manuscript accordingly (page 2, lines 8-11).

Line 31 and discussion: It is important to mention the challenges of using saliva as a specimen type for lab personnel.

We thank the reviewer for the feedback, and have revised the manuscript accordingly (page 3, lines 36-41).

Line 35: change to "upper respiratory tract".

We thank the reviewer for the feedback, and have revised the manuscript accordingly (page 3, line 35).

Line 37: "Saliva" is not a tissue. Please hazard a guess as to which tissue is responsible for saliva shedding and if it overlaps with oral and nasal swabs.

We thank the reviewer for the feedback, and have revised the manuscript accordingly (page 3, lines 42-45).

Line 42, 68: Please explain how understanding saliva shedding dynamics would impact isolation & screening, diagnostics, and treatments. This is not immediately intuitive to me.

We thank the reviewer for the feedback, and have revised the manuscript accordingly (page 3, lines 48-50).

Line 50: It would be helpful to explain why shedding duration is the best stratification variable.

We thank the reviewer for the feedback. We acknowledge that our wording was ambiguous. The clear differences in the viral dynamics patterns pertain to findings observed following the stratification, and we have revised the manuscript to make this explicit (page 4, lines 59-61).

Line 71: Dates should be listed for these studies.

We thank the reviewer for the feedback, and have revised the manuscript accordingly (page 6, lines 85-86).

**Reviewer #2 (Recommendations For The Authors):**
Please make all code and data available for replication of the analyses.

We appreciate the suggestion. Due to ethical considerations, it is not possible to make all data and code publicly available. We have clearly stated in the manuscript about it (Data availability section in Methods).

**Reviewer #3 (Recommendations For The Authors):**
Here are minor comments / technical details:(1) Figure 1B is difficult to understand.

Thank you for the comment. We updated Fig 1B to incorporate more information to aid interpretation.

(2) Did you analyse viral load or the log10 of viral load? The latter is more common. You should consider it. SI Figure 1 please plot in log10 and use a different point shape for censored data. The file quality of this figure should be improved. State in the material and methods if SE with moonlit are computed with linearization or importance sampling.

Thank you for the comment. We conducted our analyses using log10-transformed viral load. Also, we revised Supplementary Fig 1 (now renumbered as Supplementary Fig 4) as suggested. We also added Supplementary Fig 3 and clarified in the Methods that standard errors (SE) were obtained in Monolix from the Fisher information matrix using the linearization method (page 28, lines 498-499).

(3) Table 1 and Figure 3A could be collapsed.

Thank you for the comment, and we carefully considered this suggestion. Table 1 summarizes clinical variables by category, whereas Fig 3A visualizes them ordered by p-value of statistical analysis. Collapsing these into a single table would make it difficult to apprehend both the categorical summaries and the statistical ranking at a glance, thereby reducing readability. We therefore decided to retain the current layout. We appreciate the constructive feedback again.

(4) Figure 3 legend could be clarified to understand what is 3B and 3C.

We thank the reviewer for the feedback and have reinforced the description accordingly.

(5) Why use AIC instead of BICc?

Thank you for your comment. We also think BICc is a reasonable alternative. However, because our objective is predictive adequacy (reconstruction of viral dynamics), we judged AIC more appropriate. In NLMEM settings, the effective sample size required by BICc is ambiguous, making the penalty somewhat arbitrary. Moreover, since the two models reconstruct very similar dynamics, our conclusions are not sensitive to the choice of criterion.

(6) Bibliography. Most articles are with et al. (which is not standard) and some are with an extended list of names. Provide DOI for all.

We thank the reviewer for the feedback, and have revised the manuscript accordingly.

(7) Extended Table 1&2 - maybe provide a color code to better highlight some lower p-values (if you find any interesting).

We thank the reviewer for the feedback. Since no clinical information and micro-RNAs other than mir-1846 showed low p-values, we highlighted only mir-1846 with color to make it easier to locate.

(8) Please make the replication code available.

We appreciate the suggestion. Due to ethical considerations, it is not possible to make all data and code publicly available. We have clearly stated in the manuscript about it (Data availability section in Methods).